# Identifying Representations for Intervention Extrapolation

**Sorawit Saengkyongam[1]    Elan Rosenfeld[2]    Pradeep Ravikumar[2]**
**Niklas Pfister[3]    Jonas Peters[1]**

[1]ETH Zürich    [2]Carnegie Mellon University    [3]University of Copenhagen

## Abstract

The premise of identifiable and causal representation learning is to improve the current representation learning paradigm in terms of generalizability or robustness. Despite recent progress in questions of identifiability, more theoretical results demonstrating concrete advantages of these methods for downstream tasks are needed. In this paper, we consider the task of intervention extrapolation: predicting how interventions affect an outcome, even when those interventions are not observed at training time, and show that identifiable representations can provide an effective solution to this task even if the interventions affect the outcome nonlinearly. Our setup includes an outcome variable $Y$, observed features $X$, which are generated as a nonlinear transformation of latent features $Z$, and exogenous action variables $A$, which influence $Z$. The objective of intervention extrapolation is then to predict how interventions on $A$ that lie outside the training support of $A$ affect $Y$. Here, extrapolation becomes possible if the effect of $A$ on $Z$ is linear and the residual when regressing Z on A has full support. As $Z$ is latent, we combine the task of intervention extrapolation with identifiable representation learning, which we call `Rep4Ex`: we aim to map the observed features $X$ into a subspace that allows for nonlinear extrapolation in $A$. We show that the hidden representation is identifiable up to an affine transformation in $Z$-space, which, we prove, is sufficient for intervention extrapolation. The identifiability is characterized by a novel constraint describing the linearity assumption of $A$ on $Z$. Based on this insight, we propose a flexible method that enforces the linear invariance constraint and can be combined with any type of autoencoder. We validate our theoretical findings through a series of synthetic experiments and show that our approach can indeed succeed in predicting the effects of unseen interventions.

## 1 Introduction

Representation learning (see, e.g., Bengio et al., 2013, for an overview) underpins the success of modern machine learning methods as evident, for example, in their application to natural language processing and computer vision. Despite the tremendous success of such machine learning methods, it is still an open question when and to which extent they generalize to unseen data distributions. It is further unclear, which precise role representation learning can play in tackling this task.

To us, the main motivation for identifiable and causal representation learning (e.g., Schölkopf et al., 2021) is to overcome this shortcoming. The core component of this approach involves learning a representation of the data that reflects some causal aspects of the underlying model. Identifying this from the observational distribution is referred to as the identifiability problem. Without any assumptions on the data generating process, learning identifiable representations is not possible (Hyvärinen & Pajunen, 1999). To show identifiability, previous works have explored various assumptions, including the use of auxiliary information (Hyvarinen et al., 2019; Khemakhem et al., 2020), sparsity (Moran et al., 2021; Lachapelle et al., 2022), interventional data (Brehmer et al., 2022; Seigal et al., 2022; Ahuja et al., 2022a; 2023; Buchholz et al., 2023) and structural assumptions (Hälvä et al., 2021; Kivva et al., 2022). However, this body of work has focused solely on the problem of identifiability. Despite its potential, however, convincing theoretical results illustrating the benefits of such identification in solving tangible downstream tasks are arguably scarce; a few recent works have

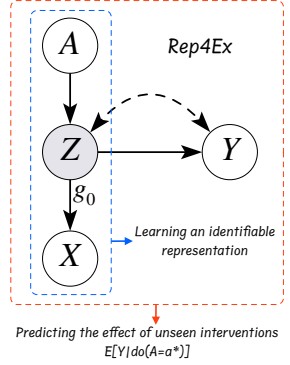

(a) Graphical model of the problem setup

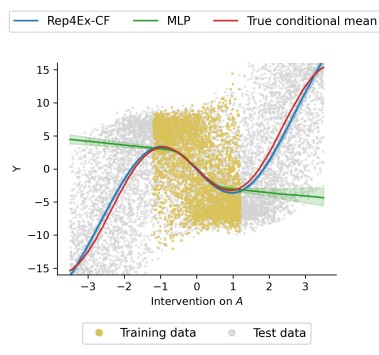

(b) Example illustrating intervention extrapolation; during training, $A$, $X$, and $Y$ are observed

Figure 1: In this paper, we consider the goal of intervention extrapolation, see (b). We are given training data (yellow) that cover only a limited range of possible values of $A$. During test time (grey), we would like to predict $\mathbb{E}[Y \mid \mathrm{do}(A = a^*)]$ for previously unseen values of $a^*$. The function $a^* \mapsto \mathbb{E}[Y \mid \mathrm{do}(A = a^*)]$ (red) can be nonlinear in $a^*$. We argue in Section 2 how this can be achieved using control functions if the data follow a structure like in (a) and $Z$ is observed. We show in Section 3 that, under suitable assumptions, the problem is still solvable if we first have to reconstruct the hidden representation $Z$ (up to a transformation) from $X$. The representation is used to predict $\mathbb{E}[Y \mid \mathrm{do}(A = a^*)]$, so we learn a representation for intervention extrapolation (Rep4Ex).

provided theoretical evidence for the advantages of identifiable representations in tasks such as estimating treatment effects (Wu & Fukumizu, 2022), improving generalization in multi-task learning (Lachapelle et al., 2023a) and generating novel object compositions (Lachapelle et al., 2023b).

In this work, we consider the task of intervention extrapolation, that is, predicting how interventions that were not present in the training data will affect an outcome. We study a setup with an outcome $Y$; observed features $X$ which are generated via nonlinear transformation of latent predictors $Z$; and exogenous action variables $A$ which influence $Z$. We assume the underlying data generating process depicted in Figure 1a. The dimension of $X$ can be larger than the dimension of $Z$ and we allow for potentially unobserved confounders between $Y$ and $Z$ (as depicted by the two-headed dotted arrow between $Z$ and $Y$). Adapting notation from the independent component analysis (ICA) literature (Hyvärinen & Oja, 2000), we refer to $g_0$ as a mixing (and $g_0^{-1}$ as an unmixing) function.

In this setup, the task of intervention extrapolation is to predict the effect of a previously unseen intervention on the action variables $A$ (with respect to the outcome $Y$). Using do-notation (Pearl, 2009), we thus aim to estimate $\mathbb{E}[Y \mid \mathrm{do}(A = a^\star)]$, where $a^\star$ lies outside the training support of $A$. Due to this extrapolation, $\mathbb{E}[Y \mid \mathrm{do}(A = a^\star)]$, which may be nonlinear in $a^\star$, cannot be consistently estimated by only considering the conditional expectation of $Y$ given $A$ (even though $A$ is exogenous and $\mathbb{E}[Y \mid \mathrm{do}(A = a)] = \mathbb{E}[Y | A = a]$ for all $a$ in the support of $A$), see Figure 1b. We formally prove this in Proposition 1. In this paper, the central assumption that permits learning identifiable representation and subsequently solving the downstream task is that the effect of $A$ on $Z$ is linear, that is, $\mathbb{E}[Z \mid A] = M_0 A$ for an unknown matrix $M_0$.

The approach we propose in this paper, Rep4Ex-CF, successfully extrapolates the effects outside the training support by performing two steps (see Figure 1a): In the first stage, we use $(A, X)$ to learn an encoder $\phi : \mathcal{X} \to \mathcal{Z}$ that identifies, from the observed distribution of $(A, X)$, the unmixing function $g_0^{-1}$ up to an affine transformation and thereby obtains a feature representation $\phi(X)$. To do that, we propose to make use of a novel constraint based on the assumption of the linear effect of $A$ on $Z$, which, as we are going to see, enables identification. Since this constraint has a simple analytical form, it can be added as a regularization term to an auto-encoder loss. In the second stage, we use $(A, \phi(X), Y)$ to estimate the interventional expression effect $\mathbb{E}[Y \mid \mathrm{do}(A = a^\star)]$. The model in the second stage is adapted from the method of control functions in the econometrics literature (Telser, 1964; Heckman, 1977; Newey et al., 1999), where one views $A$ as instrumental variables. Figure 1b shows results of our proposed method (Rep4Ex-CF) on a simulated data set, together with the outputs of and a standard neural-network-based regression (MLP).

We believe that our framework provides a complementary perspective on causal representation learning. Similar to most works in that area, we also view $Z$ as the variables that we ultimately aim to control. However, in our view, direct (or hard) interventions on $Z$ are inherently ill-defined due to its latent nature. We, therefore, consider the action variables $A$ as a means to modify the latent variables $Z$. As an example, in the context of reinforcement learning, one may view $X$ as an observable state, $Z$ as a latent state, $A$ as an action, and $Y$ as a reward. Our aim is then to identify the actions that guide us toward the desired latent state which subsequently leads to the optimal expected reward. The ability to extrapolate to unseen values of $A$ comes (partially) from the linearity of $A$ on $Z$; such extrapolation therefore becomes possible if we recover the true latent variables $Z$ up to an affine transformation. The problem of learning identifiable representations can then be understood as the process of mapping the observed features $X$ to a subspace that permits extrapolation in $A$. We refer to this task of learning a representation for intervention extrapolation as `Rep4Ex`.

## 1.1 RELATION TO EXISTING WORK

Some of the recent work on representation learning for latent causal discovery also relies on (unobserved) interventions to show identifiability, sometimes with auxiliary information. These works often assume that the interventions occur on one or a fixed group of nodes in the latent DAG (Ahuja et al., 2022a; Buchholz et al., 2023; Zhang et al., 2023) or that they are exactly paired (Brehmer et al., 2022; von Kügelgen et al., 2023). Other common conditions include parametric assumptions on the mixing function (Rosenfeld et al., 2021; Seigal et al., 2022; Ahuja et al., 2023; Varici et al., 2023) or precise structural conditions on the generative model (Cai et al., 2019; Kivva et al., 2021; Xie et al., 2022; Jiang & Aragam, 2023; Kong et al., 2023). Unlike these works, we study interventions on exogenous (or "anchor") variables, akin to simultaneous soft interventions on the latents. Identifiability is also studied in nonlinear ICA (e.g., Hyvarinen & Morioka, 2016; Hyvarinen et al., 2019; Khemakhem et al., 2020; Schell & Oberhauser, 2023), we discuss the relation in Appendix A.

The task of predicting the effects of new interventions has been explored in several prior works. Nandy et al. (2017); Saengkyongam & Silva (2020); Zhang et al. (2023) consider learning the effects of new joint interventions based on observational distribution and single interventions. Bravo-Hermsdorff et al. (2023) combine data from various regimes to predict intervention effects in previously unobserved regimes. Closely related to our work, Gultchin et al. (2021) focus on predicting causal responses for new interventions in the presence of high-dimensional mediators $X$. Unlike our work, they assume that the latent features are known and do not allow for unobserved confounders.

Our work is related to research that utilizes exogenous variables for causal effect estimation and distribution generalization. Instrumental variable (IV) approaches (Wright, 1928; Angrist et al., 1996) exploit the existence of the exogenous variables to estimate causal effects in the presence of unobserved confounders. Our work draws inspiration from the control function approach in the IV literature (Telser, 1964; Heckman, 1977; Newey et al., 1999). Several works (e.g., Rojas-Carulla et al., 2018; Arjovsky et al., 2019; Rothenhäusler et al., 2021; Christiansen et al., 2021; Rosenfeld et al., 2022; Saengkyongam et al., 2022) have used exogenous variables to increase robustness and perform distribution generalization. While the use of exogenous variables enters similarly in our approach, these existing works focus on a different task and do not allow for nonlinear extrapolation.

## 2 INTERVENTION EXTRAPOLATION WITH OBSERVED $Z$

To provide better intuition and insight into our approach, we start by considering a setup in which $Z$ is observed, which is equivalent to assuming that we are given the true underlying representation. We now focus on the intervention extrapolation part, see Figure 1a (red box) with $Z$ observed. Consider an outcome variable $Y \in \mathcal{Y} \subseteq \mathbb{R}$, predictors $Z \in \mathcal{Z} \subseteq \mathbb{R}^d$, and exogenous action variables $A \in \mathcal{A} \subseteq \mathbb{R}^k$. We assume the following structural causal model (Pearl, 2009)

$$\mathcal{S} : \quad \left\{ A := \epsilon_A \qquad Z := M_0 A + V \qquad Y := \ell(Z) + U, \right. \tag{1}$$

where $\epsilon_A, V, U$ are noise variables and we assume that $\epsilon_A \perp\!\!\!\perp (V, U)$, $\mathbb{E}[U] = 0$, and $M_0$ has full row rank. Here, $V$ and $U$ may be dependent.

**Notation.** For a structural causal model (SCM) $\mathcal{S}$, we denote by $\mathbb{P}^{\mathcal{S}}$ the observational distribution entailed by $\mathcal{S}$ and the corresponding expectation by $\mathbb{E}^{\mathcal{S}}$. When there is no ambiguity, we may omit

the superscript $\mathcal{S}$. Further, we employ the do-notation to denote the distribution and the expectation under an intervention. In particular, we write $\mathbb{P}^{\mathcal{S};\mathrm{do}(A=a)}$ and $\mathbb{E}^{S}[\cdot\,|\,\mathrm{do}(A=a)]$ to denote the distribution and the expectation under an intervention setting $A := a$, respectively, and $\mathbb{P}^{\mathrm{do}(A=a)}$ and $\mathbb{E}[\cdot\,|\,\mathrm{do}(A=a)]$ if there is no ambiguity. Lastly, for any random vector $B$, we denote by $\mathrm{supp}^{\mathcal{S}}(B)$ the support[1] of $B$ in the observational distribution $\mathbb{P}^{\mathcal{S}}$. Again, when the SCM is clear from the context, we may omit $\mathcal{S}$ and write $\mathrm{supp}(B)$ as the support in the observational distribution.

Our goal is to compute the effect of an unseen intervention on the action variables $A$ (with respect to the outcome $Y$), that is, $\mathbb{E}[Y\,|\,\mathrm{do}(A=a^{\star})]$, where $a^{\star} \notin \mathrm{supp}(A)$. A naive approach to tackle this problem is to estimate the conditional expectation $\mathbb{E}[Y|A=a]$ by regressing $Y$ on $A$ using a sample from the observational distribution of $(Y, A)$. Despite $A$ being exogenous, from (1) we only have that $\mathbb{E}[Y\,|\,\mathrm{do}(A=a)] = \mathbb{E}[Y|A=a]$ for all $a \in \mathrm{supp}(A)$. As $a^{\star}$ lies outside the support of $A$, we face the non-trivial challenge of extrapolation. The proposition below shows that in our model class $\mathbb{E}[Y\,|\,\mathrm{do}(A=a^{\star})]$ is indeed not identifiable from the conditional expectation $\mathbb{E}[Y|A]$ alone. Consequently, $\mathbb{E}[Y\,|\,\mathrm{do}(A=a^{\star})]$ cannot be consistently estimated by simply regressing $Y$ on $A$. (The result is independent of the fact whether $Z$ is observed or not and applies to the setting of unobserved $Z$ in the same way, see Section 3. Furthermore, the result still holds even when $V$ and $U$ are independent.) All proofs can be found in Appendix D.

**Proposition 1** (Regressing $Y$ on $A$ does not suffice)**.** *There exist SCMs $\mathcal{S}_1$ and $\mathcal{S}_2$ of the form* (1) *(with the same set $\mathcal{A}$) that satisfy all of the following conditions: (i)* $\mathrm{supp}^{\mathcal{S}_1}(V) = \mathrm{supp}^{\mathcal{S}_2}(V) = \mathbb{R}$*; (ii)* $\mathrm{supp}^{\mathcal{S}_1}(A) = \mathrm{supp}^{\mathcal{S}_2}(A) \neq \mathcal{A}$ *; (iii)* $\forall a \in \mathrm{supp}^{\mathcal{S}_1}(A) : \mathbb{E}^{\mathcal{S}_1}[Y|A=a] = \mathbb{E}^{\mathcal{S}_2}[Y|A=a]$*; (iv)* $\exists \mathcal{B} \subseteq \mathcal{A}$ *with positive Lebesgue measure s.t.* $\forall a \in \mathcal{B} : \mathbb{E}^{\mathcal{S}_1}[Y\,|\,\mathrm{do}(A=a)] \neq \mathbb{E}^{\mathcal{S}_2}[Y\,|\,\mathrm{do}(A=a)]$ *(the latter implies $\mathcal{B} \cap \mathrm{supp}^{\mathcal{S}_1}(A) = \varnothing$).*

Proposition 1 affirms that relying solely on the knowledge of the conditional expectation $\mathbb{E}[Y|A]$ is not sufficient to identify the effect of an intervention outside the support of $A$. It is, however, possible to incorporate additional information beyond the conditional expectation to help us identify $\mathbb{E}[Y\,|\,\mathrm{do}(A=a^{\star})]$. In particular, inspired by the method of control functions in econometrics, we propose to identify $\mathbb{E}[Y\,|\,\mathrm{do}(A=a^{\star})]$ from the observational distribution of $(A, X, Z)$ based on the following identities,

$$
\begin{aligned}
\mathbb{E}[Y\,|\,\mathrm{do}(A=a^{\star})] &= \mathbb{E}[\ell(Z)\,|\,\mathrm{do}(A=a^{\star})] + \mathbb{E}[U\,|\,\mathrm{do}(A=a^{\star})] \\
&= \mathbb{E}[\ell(M_0 a^{\star} + V)\,|\,\mathrm{do}(A=a^{\star})] + \mathbb{E}[U\,|\,\mathrm{do}(A=a^{\star})] \\
&= \mathbb{E}[\ell(M_0 a^{\star} + V)],
\end{aligned}
\tag{2}
$$

where the last equality follows from $\mathbb{E}[U] = 0$ and the fact that, for all $a^{\star} \in \mathcal{A}$, $\mathbb{P}_{U,V} = \mathbb{P}_{U,V}^{\mathrm{do}(A=a^{\star})}$. Now, since $A \perp\!\!\!\perp V$, we have $\mathbb{E}[Z\,|\,A] = M_0 A$ and $M_0$ can be identified by regressing $Z$ on $A$. $V$ is then identified with $V = Z - M_0 A$. $V$ is called a control variable and, as argued by Newey et al. (1999), for example, it can be used to identify $\ell$: defining $\lambda : v \mapsto \mathbb{E}[U|V=v]$, we have for all $z, v \in \mathrm{supp}(Z, V)$

$$
\begin{aligned}
\mathbb{E}[Y|Z=z, V=v] &= \mathbb{E}[\ell(Z) + U|Z=z, V=v] = \ell(z) + \mathbb{E}[U|Z=z, V=v] \\
&= \ell(z) + \mathbb{E}[U|V=v] = \ell(z) + \lambda(v),
\end{aligned}
\tag{3}
$$

where in the second last equality, we have used that $U \perp\!\!\!\perp Z \mid V$ (see Lemma 8 in Appendix C). In general, (3) does not suffice to identify $\ell$ (e.g., $V$ and $Z$ are not necessarily independent of each other). Only under additional assumptions, such as parametric assumptions on the function classes, $\ell$ and $\lambda$ are identifiable up to additive constants[2]. In our work, we utilize an assumption by Newey et al. (1999) that puts restrictions on the joint support of $A$ and $V$ and identifies $\ell$ on the set $M_0 \mathrm{supp}(A) + \mathrm{supp}(V)$. Since $M_0$ and $V$ are identifiable, too, this then allows us to compute, by (2), $\mathbb{E}[Y\,|\,\mathrm{do}(A=a^{\star})]$ for all $a^{\star}$ s.t. $M_0 a^{\star} + \mathrm{supp}(V) \subseteq M_0 \mathrm{supp}(A) + \mathrm{supp}(V)$; thus, $\mathrm{supp}(V) = \mathbb{R}^d$ is a sufficient condition to identify $\mathbb{E}[Y\,|\,\mathrm{do}(A=a^{\star})]$ for all $a^{\star} \in \mathcal{A}$. This support assumption, together with the additivity of $V$ in (1), is key to ensure that the nonlinear function $\ell$ can be inferred on all of $\mathbb{R}^d$, allowing for nonlinear extrapolation. Similar ideas have been used for extrapolation in a different setting and under different assumptions by Shen & Meinshausen (2023). In some applications, we may want to compute the effect of an intervention on $A$ conditioned on $Z$, that is, $\mathbb{E}[Y|Z=z, \mathrm{do}(A=a^{\star})]$. We show in Appendix C.1 that this expression is identifiable, too.

---

[1]The support of a random vector $B \in \Omega \subseteq \mathbb{R}^q$ (for some $q \in \mathbb{N}^+$) is defined as the set of all $b \in \Omega$ for which every open neighborhood of $b$ (in $\Omega$) has positive probability.

[2]The constant can be identified by using the assumption $\mathbb{E}[U] = 0$.

## 3 INTERVENTION EXTRAPOLATION VIA IDENTIFIABLE REPRESENTATIONS

Section 2 illustrates the problem of intervention extrapolation in the setting where the latent predictors $Z$ are fully observed. We now consider the setup where we do not directly observe $Z$ but instead we observe $X$ which are generated by applying a nonlinear mixing function to $Z$. Formally, consider an outcome variable $Y \in \mathcal{Y} \subseteq \mathbb{R}$, observable features $X \in \mathcal{X} \subseteq \mathbb{R}^m$, latent predictors $Z \in \mathcal{Z} = \mathbb{R}^d$, and action variables $A \in \mathcal{A} \subseteq \mathbb{R}^k$. We model the underlying data generating process by the following SCM.

**Setting 1** (Rep4Ex). *We assume the SCM*

$$\mathcal{S}: \quad \begin{cases} A := \epsilon_A & Z := M_0 A + V \\ X := g_0(Z) & Y := \ell(Z) + U, \end{cases} \tag{4}$$

*where $\epsilon_A, V, U$ are noise variables and we assume that the covariance matrix of $\epsilon_A$ is full-rank, $\epsilon_A \perp\!\!\!\perp (V, U)$, $\mathbb{E}[U] = 0$, $\mathrm{supp}(V) = \mathbb{R}^d$, and $M_0$ has full row rank (thus $k \geq d$). Further, $g_0$ and $\ell$ are measurable functions and $g_0$ is injective. In this work, we only consider interventions on $A$. For example, we do not require that the SCM models interventions on $Z$ correctly. Possible relaxations of the linearity assumption between $A$ and $Z$ and the absence of noise in $X$ are discussed in Remark 6 in Appendix B.*

Our goal is to compute $\mathbb{E}[Y \,|\, \mathrm{do}(A = a^\star)]$ for some $a^\star \notin \mathrm{supp}(A)$. As in the case of observed $Z$, the naive method of regressing $Y$ on $A$ using a non-parametric regression fails to handle the extrapolation of $a^\star$ (see Proposition 1). We, however, can incorporate additional information beyond the conditional expectation to identify $\mathbb{E}[Y \,|\, \mathrm{do}(A = a^\star)]$ through the method of control functions. From (2), we have for all $a^\star \in \mathcal{A}$ that

$$\mathbb{E}[Y \,|\, \mathrm{do}(A = a^\star)] = \mathbb{E}[\ell(M_0 a^\star + V)]. \tag{5}$$

Unlike the case where we observe $Z$, the task of identifying the unknown components on the right-hand side of (5) becomes more intricate. In what follows, we show that if we can learn an encoder $\phi : \mathcal{X} \to \mathcal{Z}$ that identifies $g_0^{-1}$ up to an affine transformation (see Definition 2 below), we can construct a procedure that identifies the right-hand side of (5) and can thus be used to predict the effect of unseen interventions on $A$.

**Definition 2** (Affine identifiability). *Assume Setting 1. An encoder $\phi : \mathcal{X} \to \mathcal{Z}$ is said to* identify $g_0^{-1}$ up to an affine transformation *(aff-identify for short) if there exists an invertible matrix $H_\phi \in \mathbb{R}^{d \times d}$ and a vector $c_\phi \in \mathbb{R}^d$ such that*

$$\forall z \in \mathcal{Z} : (\phi \circ g_0)(z) = H_\phi z + c_\phi. \tag{6}$$

*We denote by $\kappa_\phi : z \mapsto H_\phi z + c_\phi$ the corresponding affine map.*

Under Setting 1, we show an equivalent formulation of affine identifiability in Proposition 7 stressing that $Z$ can be reconstructed from $\phi(X)$.

Next, let $\phi : \mathcal{X} \to \mathcal{Z}$ be an encoder that aff-identifies $g_0^{-1}$ and $\kappa_\phi : z \mapsto H_\phi z + c_\phi$ be the corresponding affine map. From (5), we have for all $a^\star \in \mathcal{A}$ that

$$\begin{aligned}
\mathbb{E}[Y \,|\, \mathrm{do}(A = a^\star)] &= \mathbb{E}[\ell(M_0 a^\star + V)] = \mathbb{E}[(\ell \circ \kappa_\phi^{-1})(\kappa_\phi(M_0 a^\star + V))] \\
&= \mathbb{E}[(\ell \circ \kappa_\phi^{-1})(H_\phi M_0 a^\star + c_\phi + H_\phi \mathbb{E}[V] + H_\phi(V - \mathbb{E}[V]))] \\
&= \mathbb{E}[(\ell \circ \kappa_\phi^{-1})(M_\phi a^\star + q_\phi + V_\phi)], 
\end{aligned} \tag{7}$$

where we define

$$M_\phi := H_\phi M_0, \quad q_\phi := c_\phi + H_\phi \mathbb{E}[V], \text{ and } V_\phi := H_\phi(V - \mathbb{E}[V]). \tag{8}$$

We now outline how to identify the right-hand side of (7) by using the encoder $\phi$ and formalize the result in Theorem 3.

**Identifying $M_\phi$, $q_\phi$ and $V_\phi$** Using that $\phi$ aff-identifies $g_0^{-1}$, we have (almost surely) that

$$\phi(X) = (\phi \circ g_0)(Z) = H_\phi Z + c_\phi = H_\phi M_0 A + H_\phi V + c_\phi = M_\phi A + q_\phi + V_\phi. \tag{9}$$

Now, since $V_\phi \perp\!\!\!\perp A$ (following from $V \perp\!\!\!\perp A$), we can identify the pair $(M_\phi, q_\phi)$ by regressing $\phi(X)$ on $A$. The control variable $V_\phi$ can therefore be obtained as $V_\phi = \phi(X) - (M_\phi A + q_\phi)$.

**Identifying** $\ell \circ \kappa_\phi^{-1}$    Defining $\lambda_\phi : v \mapsto \mathbb{E}[U|V_\phi = v]$, we have, for all $\omega, v \in \text{supp}((\phi(X), V_\phi))$,

$$\mathbb{E}[Y|\phi(X) = \omega, V_\phi = v] \overset{(*)}{=} \mathbb{E}[Y|\kappa_\phi(Z) = \omega, V_\phi = v] = \mathbb{E}[Y|Z = \kappa_\phi^{-1}(\omega), V_\phi = v]$$

$$= \mathbb{E}[\ell(Z) + U|Z = \kappa_\phi^{-1}(\omega), V_\phi = v]$$

$$= (\ell \circ \kappa_\phi^{-1})(\omega) + \mathbb{E}[U|Z = \kappa_\phi^{-1}(\omega), V_\phi = v]$$

$$\overset{(**)}{=} (\ell \circ \kappa_\phi^{-1})(\omega) + \mathbb{E}[U|V_\phi = v] = (\ell \circ \kappa_\phi^{-1})(\omega) + \lambda_\phi(v), \qquad (10)$$

where the equality $(*)$ holds since $\phi$ aff-identifies $g_0^{-1}$ and $(**)$ holds by Lemma 9, see Appendix C. Similarly to the case in Section 2, the functions $\ell \circ \kappa_\phi^{-1}$ and $\lambda_\phi$ are identifiable (up to additive constants) under some regularity conditions on the joint support of $A$ and $V_\phi$ (Newey et al., 1999). We make this precise in the following theorem, which summarizes the deliberations from this section.

**Theorem 3.** *Assume Setting 1 and let $\phi : \mathcal{X} \to \mathcal{Z}$ be an encoder that aff-identifies $g_0^{-1}$. Further, define the optimal linear function from $A$ to $\phi(X)$ as*[3]

$$(W_\phi, \alpha_\phi) := \underset{W \in \mathbb{R}^{d \times k}, \alpha \in \mathbb{R}^d}{\operatorname{argmin}} \mathbb{E}[\|\phi(X) - (WA + \alpha)\|^2] \qquad (11)$$

*and the control variable $\tilde{V}_\phi := \phi(X) - (W_\phi A + \alpha_\phi)$. Lastly, let $\nu : \mathcal{Z} \to \mathcal{Y}$ and $\psi : \mathcal{V} \to \mathcal{Y}$ be additive regression functions such that*

$$\forall \omega, v \in \text{supp}((\phi(X), \tilde{V}_\phi)) : \mathbb{E}[Y|\phi(X) = \omega, \tilde{V}_\phi = v] = \nu(\omega) + \psi(v). \qquad (12)$$

*If $\ell, \lambda_\phi$ are differentiable and the interior of $\text{supp}(A)$ is convex, then the following statements hold*

*(i)* $\forall a^\star \in \mathcal{A} : \mathbb{E}[Y|\,\text{do}(A = a^\star)] = \mathbb{E}[\nu(W_\phi a^\star + \alpha_\phi + \tilde{V}_\phi)] - (\mathbb{E}[\nu(\phi(X))] - \mathbb{E}[Y]) \qquad (13)$

*(ii)* $\forall x \in \text{Im}(g_0), a^\star \in \mathcal{A} : \mathbb{E}[Y|X = x, \text{do}(A = a^\star)] = \nu(\phi(x)) + \psi(\phi(x) - (W_\phi a^\star + \alpha_\phi)). \qquad (14)$

# 4   IDENTIFICATION OF THE UNMIXING FUNCTION $g_0^{-1}$

Theorem 3 illustrates that intervention extrapolation can be achieved if one can identify the unmixing function $g_0^{-1}$ up to an affine transformation. In this section, we focus on the representation part (see Figure 1a, blue box) and prove that such an identification is possible. The identification relies on two key assumptions outlined in Setting 1: (i) the exogeneity of $A$ and (ii) the linearity of the effect of $A$ on $Z$. These two assumptions give rise to a conditional moment restriction on the residuals obtained from the linear regression of $g_0^{-1}(X)$ on $A$. Recall that for all encoders $\phi : \mathcal{X} \to \mathcal{Z}$ we defined $(W_\phi, \alpha_\phi) := \operatorname{argmin}_{W \in \mathbb{R}^{d \times k}, \alpha \in \mathbb{R}^d} \mathbb{E}[\|\phi(X) - (WA + \alpha)\|^2]$. Under Setting 1, we have

$$\forall a \in \text{supp}(A) : \mathbb{E}[g_0^{-1}(X) - (W_{g_0^{-1}} A + \alpha_{g_0^{-1}}) \mid A = a] = 0. \qquad (15)$$

The conditional moment restriction (15) motivates us to introduce the notion of linear invariance of an encoder $\phi$ (with respect to $A$).

**Definition 4** (Linear invariance). *Assume Setting 1. An encoder $\phi : \mathcal{X} \to \mathcal{Z}$ is said to be linearly invariant (with respect to $A$) if the following holds*

$$\forall a \in \text{supp}(A) : \mathbb{E}[\phi(X) - (W_\phi A + \alpha_\phi) \mid A = a] = 0. \qquad (16)$$

To establish identifiability, we consider an encoder $\phi : \mathcal{X} \to \mathcal{Z}$ satisfying the following constraints.

(i) $\phi$ is linearly invariant     and     (ii) $\phi|_{\text{Im}(g_0)}$ is bijective, $\qquad (17)$

where $\phi|_{\text{Im}(g_0)}$ denotes the restriction of $\phi$ to the image of the mixing function $g_0$. The second constraint (invertibility) rules out trivial solutions of the first constraint (linear invariance). For instance, a constant encoder $\phi : x \mapsto c$ (for some $c \in \mathbb{R}^d$) satisfies the linear invariance constraint but it clearly does not aff-identify $g_0^{-1}$. Theorem 5 shows that, under the assumptions listed below, the constraints (17) are necessary and sufficient conditions for an encoder $\phi$ to aff-identify $g_0^{-1}$.

---

[3]Here, $W_\phi$, $\alpha_\phi$ and $\tilde{V}_\phi$ are equal to $M_\phi$, $q_\phi$ and $V_\phi$ as shown in the proof. We introduce the new notation (e.g., (11) instead of (8)) to emphasize that the expressions are functions of the observational distribution.

**Assumption 1** (Regularity conditions on $g_0$). *Assume Setting 1. The mixing function $g_0$ is differentiable and Lipschitz continuous.*

**Assumption 2** (Regularity conditions on $V$). *Assume Setting 1. First, the characteristic function of the noise variable $V$ has no zeros. Second, the distribution $\mathbb{P}_V$ admits a density $f_V$ w.r.t. Lebesgue measure such that $f_V$ is analytic on $\mathbb{R}^d$.*

**Assumption 3** (Regularity condition on $A$). *Assume Setting 1. The support of $A$, $\mathrm{supp}(A)$, contains a non-empty open subset of $\mathbb{R}^k$.*

In addition to the injectivity assumed in Setting 1, Assumption 1 imposes further regularity conditions on the mixing function $g_0$. As for Assumption 2, the first condition is satisfied, for example, when the distribution of $V$ is infinitely divisible. The second condition requires that the density function of $V$ can be locally expressed as a convergent power series. Examples of such functions are the exponential functions, trigonometric functions, and any linear combinations, compositions, and products of those. Hence, Gaussians and mixture of Gaussians are examples of distributions that satisfy Assumption 2. Lastly, Assumption 3 imposes a condition on the support of $M_0 A$, that is, the support of $M_0 A$ has non-zero Lebesgue measure. These assumptions are closely related to the assumptions for bounded completeness in instrumental variable problems (D'Haultfoeuille, 2011).

**Theorem 5.** *Assume Setting 1 and Assumptions 1, 2, and 3. Let $\Phi$ be a class of functions from $\mathcal{X}$ to $\mathcal{Z}$ that are differentiable and Lipschitz continuous. It holds for all $\phi \in \Phi$ that*

$$\phi \text{ satisfies (17)} \iff \phi \text{ aff-identifies } g_0^{-1}. \tag{18}$$

## 5 A method for tackling Rep4Ex

### 5.1 First-stage: auto-encoder with MMR regularization

This section illustrates how to turn the identifiability result outlined in Section 4 into a practical method that implements the linear invariance and invertibility constraints in (17). The method is based on an auto-encoder (Kramer, 1991; Goodfellow et al., 2016) with a regularization term that enforces the linear invariance constraint (16). In particular, we adopt the the framework of maximum moment restrictions (MMRs) introduced in Muandet et al. (2020) as a representation of the constraint (16). MMRs can be seen as the reproducing kernel Hilbert space (RKHS) representations of conditional moment restrictions. Formally, let $\mathcal{H}$ be the RKHS of vector-valued functions (Alvarez et al., 2012) from $\mathcal{A}$ to $\mathcal{Z}$ with a reproducing kernel $k$ and define $\psi := \psi_{\mathbb{P}_{X,A}} : (x, a, \phi) \mapsto \phi(x) - (W_\phi a + \alpha_\phi)$ (recall that $W_\phi$ and $\alpha_\phi$ depend on the observational distribution $\mathbb{P}_{X,A}$). We can turn the conditional moment restriction in (16) into the MMR as follows. Define the function

$$Q(\phi) := \sup_{h \in \mathcal{H}, \|h\| \leq 1} \left( \mathbb{E}[\psi(X, A, \phi)^\top h(A)] \right)^2. \tag{19}$$

If the reproducing kernel $k$ is integrally strictly positive definite (see Muandet et al. (2020, Definition 2.1)), then $Q(\phi) = 0$ if and only if the conditional moment restriction in (16) is satisfied.

One of the main advantages of using the MMR representation is that it can be written as a closed-form expression. We have by Muandet et al. (2020, Theorem 3.3) that

$$Q(\phi) = \mathbb{E}[\psi(X, A, \phi)^\top k(A, A')\psi(X', A', \phi)], \tag{20}$$

where $(X', A')$ is an independent copy of $(X, A)$.

We now introduce our auto-encoder objective function[4] with the MMR regularization. Let $\phi : \mathcal{X} \to \mathcal{Z}$ be an encoder and $\eta : \mathcal{Z} \mapsto \mathcal{X}$ be a decoder. Our (population) loss function is defined as

$$\mathcal{L}(\phi, \eta) := \mathbb{E}[\|X - \eta(\phi(X))\|^2] + \lambda Q(\phi), \tag{21}$$

where $\lambda$ is a regularization parameter. In practice, we parameterize $\phi$ and $\eta$ by neural networks, use a plug-in estimator[5] for (21) to obtain an empirical loss function, and minimize that loss with a

---

[4]We consider a basic auto-encoder, but one can add MMR regularization to other variants too, e.g., (Kingma & Welling, 2014), adversarial-based (Makhzani et al., 2015), or diffusion-based (Preechakul et al., 2022).

[5]More precisely, we replace the expectations in (21) and (11) by empirical means (the latter expression enters through $\psi$ and $Q(\phi)$).

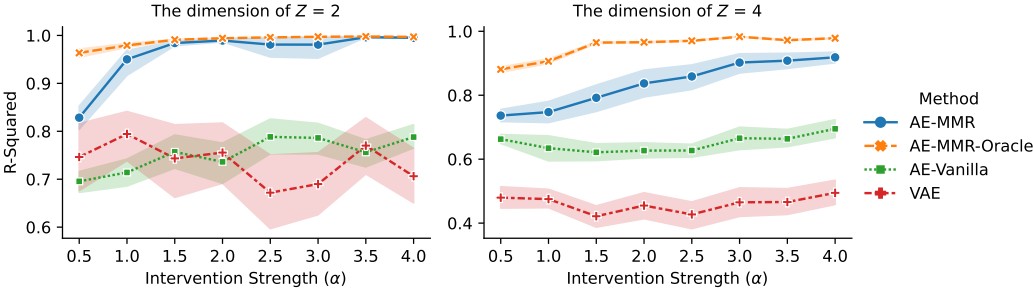

Figure 2: R-squared values for different methods as the intervention strength ($\alpha$) increases. Each point represents an average over 20 repetitions, and the error bar indicates its 95% confidence interval. AE-MMR yields an R-squared close to 1 as $\alpha$ increases, indicating its ability to aff-identify $g_0^{-1}$, while the two baseline methods yield significantly lower R-squared values.

standard (stochastic) gradient descent optimizer. Here, the role of the reconstruction loss part in (21) is to enforce the bijectivity constraint of $\phi|_{\text{Im}(g_0)}$ in (17). The regularization parameter $\lambda$ controls the trade-off between minimizing the mean squared error (MSE) and satisfying the MMR. We discuss procedures to choose $\lambda$ in Appendix E.2.

## 5.2 SECOND-STAGE: CONTROL FUNCTION APPROACH

Given a learned encoder $\phi$, we can now implement the control function approach for estimating $\mathbb{E}[Y \mid \text{do}(A = a^\star)]$, as per Theorem 3. We call the procedure Rep4Ex-CF. Algorithm 1 in Appendix E outlines the details. In summary, we first perform the linear regression of $\phi(X)$ on $A$ to obtain $(\hat{W}_\phi, \hat{\alpha}_\phi)$, allowing us to compute the control variables $\hat{V} = \phi(X) - (\hat{W}_\phi A - \hat{\alpha}_\phi)$. Subsequently, we employ an additive regression model on $(\phi(X), \hat{V})$ to predict $Y$ and obtain the additive regression functions $\hat{\nu}$ and $\hat{\psi}$. Finally, using the function $\hat{\nu}$, we compute an empirical average of the expectation on the right-hand side of (13).

## 6 EXPERIMENTS

We now conduct simulation experiments to empirically validate our theoretical findings. First, we apply the MMR based auto-encoder introduced in Section 5.1 and show in Section 6.1 that it can successfully recover the unmixing function $g_0^{-1}$ up to an affine transformation. Second, in Section 6.2, we apply the full Rep4Ex-CF procedure (see Section 5.2) to demonstrate that one can indeed predict previously unseen interventions as suggested by Theorem 3. The code for all experiments is included in the supplementary material.

## 6.1 IDENTIFYING THE UNMIXING FUNCTION $g_0^{-1}$

This section validates the result of affine identifiability , see Theorem 5. We consider the SCMs

$$\mathcal{S}(\alpha): \quad \{A \coloneqq \epsilon_A \qquad Z \coloneqq \alpha M_0 A + V \qquad X \coloneqq g_0(Z), \tag{22}$$

where the complete specification of this SCM is given in Appendix G.1. The parameter $\alpha$ controls the strength of the effect of $A$ on $Z$. We set the dimension of $X$ to 10 and consider two choices $d \in \{2, 4\}$ for the dimension of $Z$. Additionally, we set the dimension of $A$ to the dimension of $Z$.

We sample 1'000 observations from the SCM (22) and learn an encoder $\phi$ using the regularized auto-encoder (AE-MMR) as outlined in Section 5.1. As our baselines, we include a vanilla auto-encoder (AE-Vanilla) and a variational auto-encoder (VAE) for comparison. We also consider an oracle model (AE-MMR-Oracle) where we train the encoder and decoder using the true latent predictors $Z$ and then use these trained models to initialize the regularized auto-encoder. We refer to Appendix G.2 for the details on the network and parameter choices. Lastly, we consider identifiability of $g_0^{-1}$ only up to an affine transformation, see Definition 2. To measure the quality of

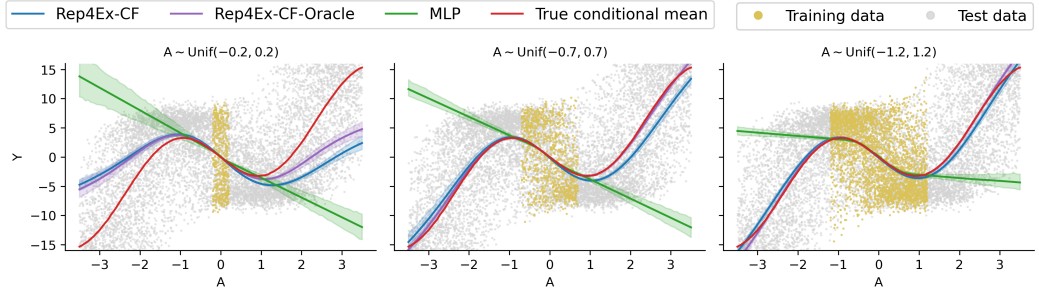

Figure 3: Different estimations of the target of inference $\mathbb{E}[Y \mid \mathrm{do}(A := \cdot)]$ as the training support $\gamma$ increases. The error bars represent the 95% confidence intervals over 10 repetitions. The training points displayed are subsampled for the purpose of visualization. `Rep4Ex-CF` demonstrates the ability to extrapolate beyond the training support, achieving nearly perfect extrapolation when $\gamma = 1.2$. In contrast, the baseline `MLP` shows clear limitations in its ability to extrapolate.

an estimate $\phi$, we therefore linearly regress the true $Z$ on the representation $\phi(X)$ and report the R-squared for each candidate method. This metric is justified by Proposition 7 in Appendix C.2.

Figure 2 illustrates the results with varying intervention strength ($\alpha$). As $\alpha$ increases, our method, `AE-MMR`, achieves higher R-squared values that appear to approach 1. This indicates that `AE-MMR` can indeed recover the unmixing function $g_0^{-1}$ up to an affine transformation. In contrast, the two baseline methods, `AE-Vanilla` and `VAE`, achieve significantly lower R-squared values, indicating non-identifiablity without enforcing the linear invariance constraint, see also the scatter plots in Figures 5 (`AE-MMR`) and 6 (`AE-Vanilla`) in Appendix H.

### 6.2 PREDICTING PREVIOUSLY UNSEEN INTERVENTIONS

In this section, we focus on the task of predicting previously unseen interventions as detailed in Section 3. We use the following SCM as data generating process.

$$\mathcal{S}(\gamma): \quad \left\{ A := \epsilon_A^\gamma \qquad Z := M_0 A + V \qquad X := g_0(Z) \qquad Y := \ell(Z) + U, \right. \tag{23}$$

where $\epsilon_A^\gamma \sim \mathrm{Unif}([-\gamma, \gamma]^k)$. Hence, the parameter $\gamma$ determines the support of $A$ in the observational distribution. The complete specification of this SCM is provided in Appendix G.1.

Our approach, denoted by `Rep4Ex-CF`, follows the procedure outlined in Algorithm 1. In the first stage, we employ `AE-MMR` as the regularized auto-encoder. In the second stage, we use a neural network that enforces additivity in the output layer for the additive regression model. For comparison, we include a neural-network-based regression model (`MLP`) of $Y$ on $A$ as a baseline. We also include an oracle method, `Rep4Ex-CF-Oracle`, where we use the true latent $Z$ instead of learning a representation in the first stage. In all experiments, we use a sample size of 10'000.

Figure 3 presents the results obtained with three $\gamma$ values (0.2, 0.7, 1.2), one-dimensional $A$ and two-dimensional $X$. As anticipated, the neural-network-based regression model (`MLP`) fails to extrapolate beyond the training support. Conversely, our approach, `Rep4Ex-CF`, demonstrates successful extrapolation, with increased performance for higher $\gamma$. Furthermore, we conduct experiments with multi-dimensional $A$ and present the results in Appendix H.1. Solving the optimization problem becomes more difficult but the outcomes echo the results observed with one-dimensional $A$.

## 7 DISCUSSION

Our work highlights concrete benefits of identifiable representation learning. We introduce `Rep4Ex`, the task of learning a representation that enables nonlinear intervention extrapolation and propose corresponding theory and methodology. We regard this work only as a first step toward solving this task. Developing alternative methods and relaxing some of the assumptions (e.g., allowing for noise in the mixing function $g_0$ and more flexible dependencies between $A$ and $Z$) may yield more powerful methods for achieving `Rep4Ex`.

## ACKNOWLEDGMENTS

We thank Nicola Gnecco and Felix Schur for helpful discussions. NP was supported by a research grant (0069071) from Novo Nordisk Fonden. ER and PR acknowledge the support of DARPA via FA8750-23-2-1015, ONR via N00014-23-1-2368, and NSF via IIS-1909816, IIS-1955532.

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

# Appendices

## A   RELATED WORK: NONLINEAR ICA

Identifiable representation learning has been studied within the framework of nonlinear ICA (e.g., Hyvarinen & Morioka, 2016; Hyvarinen et al., 2019; Khemakhem et al., 2020; Schell & Oberhauser, 2023). Khemakhem et al. (2020) provide a unifying framework that leverages the independence structure of latent variables $Z$ conditioned on auxiliary variables. Although our actions $A$ could be considered auxiliary variables, the identifiability results and assumptions in Khemakhem et al. (2020) do not fit our setup and task. Concretely, a key assumption in their framework is that the components of $Z$ are independent when conditioned on $A$. In contrast, our approach permits dependence among the components of $Z$ even when conditioned on $A$ as the components of $V$ in our setting can have arbitrary dependencies. More importantly, Khemakhem et al. (2020) provide identifiability up to point-wise nonlinearities which is not sufficient for intervention extrapolation. The main focus of our work is to provide an identification that facilitates a solution to the task of intervention extrapolation. Some other studies in nonlinear ICA have shown identifiability beyond point-wise nonlinearities (e.g., Roeder et al., 2021; Ahuja et al., 2022b). However, the models considered in these studies are not compatible with our data generation process either.

## B   REMARK ON THE KEY ASSUMPTIONS IN SETTING 1

**Remark 6.** *(i) The assumption of linearity from $Z$ on $A$ can be relaxed: if there is a known nonlinear function $h$ such that $Z := M_0 h(\tilde{A}) + V$, we can define $A := h(\tilde{A})$ and obtain an instance of Setting 1. Similarly, if there is an injective $h$ such that $\tilde{Z} := h(M_0 A + V)$ and $X := g_0(\tilde{Z})$, we can define $Z := M_0 A + V$ and $X := (g_0 \circ h)(Z)$. (ii) The assumptions of full support of $V$ and full rank of $M_0$ can be relaxed by considering $\mathcal{Z} \subseteq \mathbb{R}^d$ to be a linear subspace, with $\operatorname{supp}(V)$ and $M_0 \mathcal{A}$ both being equal to $\mathcal{Z}$. (iii) Our experimental results in Appendix H.3 suggest that it may be possible to relax the assumption of the absence of noise in $X$.*

## C   FURTHER THEORETICAL RESULTS

### C.1   IDENTIFYING $\mathbb{E}[Y|Z = z, \operatorname{do}(A = a^\star)]$ WITH OBSERVED $Z$

For all $z \in \operatorname{supp}(Z)$ and $a^\star \in \mathcal{A}$, we have

$$
\begin{aligned}
\mathbb{E}[Y|Z = z, \operatorname{do}(A = a^\star)] &= \ell(z) + \mathbb{E}[U|Z = z, \operatorname{do}(A = a^\star)] \\
&= \ell(z) + \mathbb{E}[U|M_0 a^\star + V = z, \operatorname{do}(A = a^\star)] \\
&= \ell(z) + \mathbb{E}[U|V = z - M_0 a^\star, \operatorname{do}(A = a^\star)] \\
&= \ell(z) + \mathbb{E}[U|V = z - M_0 a^\star] \qquad \text{since } \mathbb{P}_{U,V} = \mathbb{P}_{U,V}^{\operatorname{do}(A=a^\star)}, \\
&= \ell(z) + \lambda(z - M_0 a^\star),
\end{aligned}
$$

where, $\ell$ and $\lambda$ are identifiable by (3) if the interior of $\operatorname{supp}(A)$ is convex (we still assume that $\operatorname{supp}(V) = \mathbb{R}^d$), and the functions $\ell$ and $\lambda$ are differentiable (Newey et al., 1999).

### C.2   EQUIVALENT FORMULATION OF AFFINE IDENTIFIABILITY

Under Setting 1, we show an equivalent formulation of affine identifiability in Proposition 7 stressing that $Z$ can be reconstructed from $\phi(X)$. In our empirical evaluation (see Section 6), we adopt this formulation to define a metric for measuring how well an encoder $\phi$ aff-identifies $g_0^{-1}$.

**Proposition 7** (Equivalent definition of affine identifiability)**.** *Assume Setting 1. An encoder $\phi : \mathcal{X} \to \mathcal{Z}$ aff-identifies $g_0^{-1}$ if and only if there exists a matrix $J_\phi \in \mathbb{R}^{d \times d}$ and a vector $d_\phi \in \mathbb{R}^d$ s.t.*

$$
\forall z \in \mathcal{Z} : z = J_\phi \phi(x) + d_\phi, \quad \text{where } x := g_0(z). \tag{24}
$$

### C.3 SOME LEMMATA

**Lemma 8.** *Assume the underlying SCM* (1). *We have that $U \perp\!\!\!\perp Z \mid V$ under $\mathbb{P}^{\mathcal{S}}$.*

*Proof.* In the SCM (1) it holds that $A \perp\!\!\!\perp (U, V)$ which by the weak union property of conditional independence (e.g., Constantinou & Dawid, 2017, Theorem 2.4) implies that $A \perp\!\!\!\perp U \mid V$. This in turn implies $(A, V) \perp\!\!\!\perp U \mid V$ (e.g., Constantinou & Dawid, 2017, Example 2.1). Now, by Proposition 2.3 (ii) in Constantinou & Dawid (2017) this is equivalent to the condition that for all measurable and bounded functions $g : \mathcal{A} \times \mathbb{R} :\to \mathbb{R}$ it almost surely holds that

$$\mathbb{E}[g(A, V) \mid U, V] = \mathbb{E}[g(A, V) \mid V]. \tag{25}$$

Hence, for all $f : \mathcal{Z} \to \mathbb{R}$ measurable and bounded it almost surely holds that

$$
\begin{aligned}
\mathbb{E}[f(Z) \mid U, V] &= \mathbb{E}[f(M_0 A + V) \mid U, V] \\
&= \mathbb{E}[f(M_0 A + V) \mid V] \qquad \text{by (25) with } g : (a, v) \mapsto f(M_0 a + v) \\
&= \mathbb{E}[f(Z) \mid V].
\end{aligned}
\tag{26}
$$

Again by Proposition 2.3 (ii) in Constantinou & Dawid (2017), this is equivalent to $U \perp\!\!\!\perp Z \mid V$ as desired. $\qquad\square$

As an alternative to our proof, one can also argue using SCMs and Markov properties in ADMGs (Richardson, 2003).

**Lemma 9.** *Assume Setting 1. We have that $U \perp\!\!\!\perp Z \mid V_\phi$.*

*Proof.* Since the function $v \mapsto H_\phi(v - \mathbb{E}[V])$ is bijective, the proof follows from the same arguments as given in the proof of Lemma 8. $\qquad\square$

**Lemma 10.** *Assume Setting 1. Let $\phi : \mathcal{X} \to \mathcal{Z}$ be an encoder. We have that*

$$\phi \circ g_0 \text{ is bijective} \implies \phi|_{\mathrm{Im}(g_0)} \text{ is bijective} . \tag{27}$$

*Proof.* Let $\phi$ be an encoder such that $\phi \circ g_0$ is bijective. We first show that $\phi|_{\mathrm{Im}(g_0)}$ is injective by contradiction. Assume that $\phi|_{\mathrm{Im}(g_0)}$ is not injective. Then, there exist $x_1, x_2 \in \mathrm{Im}(g_0)$ such that $\phi(x_1) = \phi(x_2)$ and $x_1 \neq x_2$. Now consider $z_1, z_2 \in \mathcal{Z}$ with $x_1 = g_0(z_1)$ and $x_2 = g_0(z_2)$; clearly, $z_1 \neq z_2$. Using that $\phi \circ g_0$ is injective, we have $(\phi \circ g_0)(z_1) = \phi(x_1) \neq \phi(x_2) = (\phi \circ g_0)(z_2)$ which leads to the contradiction. We can thus conclude that $\phi|_{\mathrm{Im}(g_0)}$ is injective.

Next, we show that $\phi|_{\mathrm{Im}(g_0)}$ is surjective. Let $z_1, z_2 \in \mathcal{Z}$. Since $\phi \circ g_0$ is surjective, there exist $\tilde{z}_1, \tilde{z}_2 \in \mathcal{Z}$ such that $z_1 = (\phi \circ g_0)(\tilde{z}_1)$ and $z_2 = (\phi \circ g_0)(\tilde{z}_2)$. Let $x_1 := g_0(\tilde{z}_1) \in \mathrm{Im}(g_0)$ and $x_2 := g_0(\tilde{z}_2) \in \mathrm{Im}(g_0)$. We then have that $z_1 = \phi(x_1)$ and $z_2 = \phi(x_2)$ which shows that $\phi|_{\mathrm{Im}(g_0)}$ is surjective and concludes the proof. $\qquad\square$

## D PROOFS

### D.1 PROOF OF PROPOSITION 1

*Proof.* We consider $k = d = 1$, that is, $A \in \mathbb{R}, Z \in \mathbb{R}, Y \in \mathbb{R}$. We define the function $p_V^1 : \mathbb{R} \to \mathbb{R}$ for all $v \in \mathbb{R}$ by

$$
p_V^1(v) = \begin{cases} \frac{1}{12} & \text{if } v \in (-4, 2) \\ \frac{1}{4} \exp(-(v - 2)) & \text{if } v \in (2, \infty) \\ \frac{1}{4} \exp(v + 4) & \text{if } v \in (-\infty, -4) \end{cases}
$$

and the function $p_V^2 : \mathbb{R} \to \mathbb{R}$ for all $v \in \mathbb{R}$ by

$$
p_V^2(v) = \begin{cases} \frac{1}{12} & \text{if } v \in (-2, 1) \\ \frac{1}{24} & \text{if } v \in (-5, -2) \\ \frac{5}{16} \exp(-(v - 1)) & \text{if } v \in (1, \infty) \\ \frac{5}{16} \exp(v + 5) & \text{if } v \in (-\infty, -5). \end{cases}
$$

These two functions are valid densities as we have for all $v \in \mathbb{R}$ that $p_V^1(v) > 0$, $\forall v \in \mathbb{R} : p_V^2(v) > 0$, and $\int_{-\infty}^{\infty} p_V^1(v)\, dv = 1$, $\int_{-\infty}^{\infty} p_V^2(v)\, dv = 1$. Furthermore, these two densities $p_V^1(v)$ and $p_V^2(v)$ satisfy the following conditions,

(1) for all $a \in (0,1)$, it holds that

$$\int_{a-1}^{a+1} p_V^1(v)\, dv = \frac{1}{6} = \int_{a-2}^{a} p_V^2(v)\, dv, \tag{28}$$

(2) for all $a \in (-3,-2)$ the following holds

$$
\begin{aligned}
\int_{a-1}^{a+1} p_V^1(v)\, dv &= \int_{a-1}^{a+1} \frac{1}{4} \exp(v+4)\, dv \\
&\geq \frac{1}{2} \exp((a-1)+4) \\
&\geq \frac{1}{2} \\
&> \frac{1}{12} \\
&= \int_{a-2}^{a} p_V^2(v)\, dv. \tag{29}
\end{aligned}
$$

Next, let $\mathcal{S}_1$ be the following SCM

$$\mathcal{S}_1 : \quad \begin{cases} A := \epsilon_A \\ Z := -A + V \\ Y := \mathbb{1}(|Z| \leq 1) + U, \end{cases} \tag{30}$$

where $\epsilon_A \sim \text{Uniform}(0,1)$, $V \sim \mathbb{P}_V^1$, $U \sim \mathbb{P}_U^1$ independent such that $\epsilon_A \perp\!\!\!\perp (V,U)$, and $\mathbb{E}[U] = 0$. Further, we assume that $V$ admits a density $p_V^1$ as defined above.

Next, we define the second SCM $\mathcal{S}_2$ as follows

$$\mathcal{S}_2 : \quad \begin{cases} A := \epsilon_A \\ Z := -A + V \\ Y := \mathbb{1}(|Z+1| \leq 1) + U, \end{cases} \tag{31}$$

where $\epsilon_A \sim \text{Uniform}(0,1)$, $V \sim \mathbb{P}_V^2$, $U \sim \mathbb{P}_U^2$ independent such that $\epsilon_A \perp\!\!\!\perp (V,U)$, $\mathbb{E}[U] = 0$ and $V$ has the density given by $p_V^2$. By construction we have that $\text{supp}^{\mathcal{S}_1}(V) = \text{supp}^{\mathcal{S}_2}(V) = \mathbb{R}$ and $\text{supp}^{\mathcal{S}_1}(A) = \text{supp}^{\mathcal{S}_2}(A)$. Now, we show that the two SCMs $\mathcal{S}_1$ and $\mathcal{S}_2$ satisfy the third statement of Proposition 1. Define $c_1 = 0$ and $c_2 = 1$. For $i \in \{1,2\}$, we have for all $a \in \mathbb{R}$ that

$$
\begin{aligned}
\mathbb{E}^{\mathcal{S}_i}[Y \mid \text{do}(A=a)] &= \mathbb{E}^{\mathcal{S}_i}[\mathbb{1}(|Z+c_i| \leq 1) \mid \text{do}(A=a)] + \mathbb{E}^{\mathcal{S}_i}[U \mid \text{do}(A=a)] \\
&= \mathbb{E}^{\mathcal{S}_i}[\mathbb{1}(|V-a+c_i| \leq 1) \mid \text{do}(A=a)] + \mathbb{E}^{\mathcal{S}_i}[U \mid \text{do}(A=a)] \\
&\overset{(*)}{=} \mathbb{E}^{\mathcal{S}_i}[\mathbb{1}(|V-a+c_i| \leq 1) \mid \text{do}(A=a)] \\
&\overset{(**)}{=} \mathbb{E}^{\mathcal{S}_i}[\mathbb{1}(|V-a+c_i| \leq 1)] \\
&= \int_{-\infty}^{\infty} \mathbb{1}(|v-a+c_i| \leq 1) p_V^i(v)\, dv, \tag{32}
\end{aligned}
$$

where $(*)$ holds because $\forall a \in \mathcal{A} : \mathbb{P}_U = \mathbb{P}_U^{\text{do}(A=a)}$ and $\mathbb{E}^{\mathcal{S}_i}[U] = 0$ and $(**)$ holds because $\forall a \in \mathcal{A} : \mathbb{P}_V = \mathbb{P}_V^{\text{do}(A=a)}$. Since $A$ is exogenous, we have for all $i \in \{1,2\}$ and $a \in \text{supp}^{\mathcal{S}_1}(A) = (0,1)$

that $\mathbb{E}^{\mathcal{S}_i}[Y \mid \mathrm{do}(A = a)] = \mathbb{E}^{\mathcal{S}_i}[Y \mid A = a]$. From (32), we therefore have for all $a \in (0, 1)$

$$
\begin{aligned}
\mathbb{E}^{\mathcal{S}_1}[Y \mid A = a] &= \int_{-\infty}^{\infty} \mathbb{1}(|v - a| \le 1) p_V^1(v)\, dv \\
&= \int_{a-1}^{a+1} p_V^1(v)\, dv \\
&= \int_{a-2}^{a} p_V^2(v)\, dv \qquad\qquad \text{by (28)} \\
&= \int_{-\infty}^{\infty} \mathbb{1}(|v - a + 1| \le 1) p_V^2(v)\, dv \\
&= \mathbb{E}^{\mathcal{S}_2}[Y \mid A = a].
\end{aligned}
$$

We have shown that the two SCMs $\mathcal{S}_1$ and $\mathcal{S}_2$ satisfy the first statement of Proposition 1. Lastly, we show below that they also satisfy the fourth statement of Proposition 1. Define $\mathcal{B} := (-3, -2) \subseteq \mathbb{R}$ which has positive measure. From (32), we then have for all $a \in (-3, -2)$

$$
\begin{aligned}
\mathbb{E}^{\mathcal{S}_1}[Y \mid \mathrm{do}(A = a)] &= \int_{-\infty}^{\infty} \mathbb{1}(|v - a| \le 1) p_V^1(v)\, dv \\
&= \int_{a-1}^{a+1} p_V^1(v)\, dv \\
&\ne \int_{a-2}^{a} p_V^2(v)\, dv \qquad\qquad \text{by (29)} \\
&= \int_{-\infty}^{\infty} \mathbb{1}(|v - a + 1| \le 1) p_V^2(v)\, dv \\
&= \mathbb{E}^{\mathcal{S}_2}[Y \mid \mathrm{do}(A = a)],
\end{aligned}
$$

which shows that $\mathcal{S}_1$ and $\mathcal{S}_2$ satisfy the forth condition of Proposition 1 and concludes the proof. $\quad\square$

### D.2 PROOF OF PROPOSITION 7

*Proof.* We begin by showing the 'only if' direction. Let $\phi : \mathcal{X} \to \mathcal{Z}$ be an encoder that aff-identifies $g_0^{-1}$. Then, by definition, there exists an invertible matrix $H_\phi \in \mathbb{R}^{d \times d}$ and a vector $c_\phi \in \mathbb{R}^d$ such that

$$
\forall z \in \mathcal{Z} : (\phi \circ g_0)(z) = H_\phi z + c_\phi. \tag{33}
$$

We then have that

$$
\forall z \in \mathcal{Z} : z = H_\phi^{-1} \phi(x) - H_\phi^{-1} c_\phi, \quad \text{where } x := g_0(z), \tag{34}
$$

which shows the required statement.

Next, we show the 'if' direction. Let $\phi : \mathcal{X} \to \mathcal{Z}$ be an encoder for which there exists a matrix $J_\phi \in \mathbb{R}^{d \times d}$ and a vector $d_\phi \in \mathbb{R}^d$ such that

$$
\forall z \in \mathcal{Z} : z = J_\phi \phi(x) + d_\phi, \quad \text{where } x := g_0(z). \tag{35}
$$

Since $\mathcal{Z} = \mathbb{R}^d$, this implies that $J_\phi$ is surjective and thus has full rank. We therefore have that

$$
\forall z \in \mathcal{Z} : (\phi \circ g_0)(z) = J_\phi^{-1} z - J_\phi^{-1} d_\phi, \tag{36}
$$

which shows the required statement and concludes the proof.

$\square$

### D.3 PROOF OF THEOREM 3

*Proof.* Let $\kappa_\phi = z \mapsto H_\phi z + c_\phi$ be the corresponding affine map of $\phi$. From (7), we have for all $a^\star \in \mathcal{A}$, that

$$
\mathbb{E}[Y \mid \mathrm{do}(A = a^\star)] = \mathbb{E}[(\ell \circ \kappa_\phi^{-1})(M_\phi a^\star + q_\phi + V_\phi)], \tag{37}
$$

where $M_\phi = H_\phi M_0$, $q_\phi = c_\phi + H_\phi \mathbb{E}[V]$, and $V_\phi = H_\phi(V - \mathbb{E}[V])$ as defined in (8). To prove the first statement, we thus aim to show that, for all $a^\star \in \mathcal{A}$,

$$\mathbb{E}[\nu(W_\phi a^\star + \alpha_\phi + \tilde{V}_\phi)] - (\mathbb{E}[\nu(\phi(X))] - \mathbb{E}[Y]) = \mathbb{E}[(\ell \circ \kappa_\phi^{-1})(M_\phi a^\star + q_\phi + V_\phi)]. \qquad (38)$$

To begin with, we show that $W_\phi = M_\phi$ and $\alpha_\phi = q_\phi$. We have for all $\alpha \in \mathbb{R}^d, W \in \mathbb{R}^{d \times d}$

$$\begin{aligned}
&\mathbb{E}[\|\phi(X) - (WA + \alpha)\|^2] \\
&= \mathbb{E}[\|M_\phi A + q_\phi + V_\phi - \alpha - WA\|^2] && \text{from (9)} \\
&= \mathbb{E}[\|(M_\phi - W)A + (q_\phi - \alpha) + V_\phi\|^2] \\
&= \mathbb{E}[\|(M_\phi - W)A + (q_\phi - \alpha)\|^2] \\
&\qquad + 2\mathbb{E}[((M_\phi - W)A + (q_\phi - \alpha))^\top V_\phi] + \mathbb{E}[\|V_\phi\|^2] \\
&= \mathbb{E}[\|(M_\phi - W)A + (q_\phi - \alpha)\|^2] + \mathbb{E}[\|V_\phi\|^2]. && \text{since } A \perp\!\!\!\perp V_\phi \text{ and } \mathbb{E}[V_\phi] = 0
\end{aligned}$$

Since the covariance matrix of $A$ has full rank, we therefore have that

$$(\alpha_\phi, W_\phi) = \operatorname*{argmin}_{\alpha \in \mathbb{R}^d, W \in \mathbb{R}^{d \times k}} \mathbb{E}[\|\phi(X) - \alpha - WA\|^2] = (q_\phi, M_\phi), \qquad (39)$$

and that $\tilde{V}_\phi = \phi(X) - (M_\phi A + q_\phi) = V_\phi$, where the last equality holds by (9).

Next, we show that $\nu \equiv (\ell \circ \kappa_\phi^{-1})$. Since $\ell$ is differentiable, the function $\ell \circ \kappa_\phi^{-1}$ is also differentiable. We have $\operatorname{supp}(A, V_\phi) = \operatorname{supp}(A, V) = \operatorname{supp}(A) \times \mathbb{R}^d$. Thus, the interior of $\operatorname{supp}(A, V_\phi)$ is convex (as the interior of $\operatorname{supp}(A)$ is convex) and its boundary has measure zero. Also, the matrix $M_0$ has full row rank. Moreover, using aff-identifiability and (4) we can write

$$\phi(X) = M_\phi A + q_\phi + V_\phi$$
$$Y = \ell \circ \kappa_\phi^{-1}(\phi(X)) + U,$$

where $A \perp\!\!\!\perp (V_\phi, U)$. This is a simultaneous equation model (over the observed variables $\phi(X)$, $A$, and $Y$) for which the structural function is $\ell \circ \kappa_\phi^{-1}$ and the control function is $\lambda_\phi$. We can therefore apply Theorem 2.3 in Newey et al. (1999) (see Gnecco et al. (2023, Proposition 3) for a complete proof, including usage of convexity, which we believe is missing in the argument of Newey et al. (1999)) to conclude that $\ell \circ \kappa_\phi^{-1}$ and $\lambda_\phi$ are identifiable from (10) up to a constant. That is,

$$\nu \equiv (\ell \circ \kappa_\phi^{-1}) + \delta \qquad \text{and} \qquad \psi \equiv \lambda_\phi - \delta \qquad (40)$$

for some constant $\delta \in \mathbb{R}$. Combining with the fact that $W_\phi = M_\phi$ and $\alpha_\phi = q_\phi$, we then have, for all $a^\star \in \mathcal{A}$,

$$\mathbb{E}[\nu(W_\phi a^\star + \alpha_\phi + \tilde{V}_\phi)] = \mathbb{E}[(\ell \circ \kappa_\phi^{-1})(M_\phi a^\star + q_\phi + V_\phi)] + \delta. \qquad (41)$$

Now, we use the assumption that $\mathbb{E}[U] = 0$ to deal with the constant term $\delta$.

$$\begin{aligned}
\mathbb{E}[Y] &= \mathbb{E}[\ell(g_0^{-1}(X))] && \text{since } \mathbb{E}[U] = 0 && (42) \\
&= \mathbb{E}[((\ell \circ \kappa_\phi^{-1}) \circ (\kappa_\phi \circ g_0^{-1}))(X)] && && (43) \\
&= \mathbb{E}[(\ell \circ \kappa_\phi^{-1})(\phi(X))] && \text{since } \phi \text{ aff-identifies } g_0^{-1}. && (44)
\end{aligned}$$

Thus, we have

$$\begin{aligned}
\mathbb{E}[\nu(\phi(X))] - \mathbb{E}[Y] &= \mathbb{E}[(\ell \circ \kappa_\phi^{-1})(\phi(X)) + \delta] - \mathbb{E}[Y] && \text{by (40)} \\
&= \mathbb{E}[(\ell \circ \kappa_\phi^{-1})(\phi(X)) + \delta] - \mathbb{E}[(\ell \circ \kappa_\phi^{-1})(\phi(X))] && \text{by (44)} \\
&= \delta. && (45)
\end{aligned}$$

Combining (45) and (41), we have for all $a^\star \in \mathcal{A}$ that

$$\mathbb{E}[\nu(W_\phi a^\star + \alpha_\phi + \tilde{V}_\phi)] - (\mathbb{E}[\nu(\phi(X))] - \mathbb{E}[Y]) = \mathbb{E}[(\ell \circ \kappa_\phi^{-1})(M_\phi a^\star + q_\phi + V_\phi)],$$

which yields (38) and concludes the proof of the first statement.

Next, we prove the second statement. We have for all $x \in \text{Im}(g_0)$ and $a^\star \in \mathcal{A}$, that

$$
\begin{aligned}
\mathbb{E}[Y|X = x, \text{do}(A = a^\star)] &= \mathbb{E}[\ell(Z) \mid X = x, \text{do}(A = a^\star)] + \mathbb{E}[U|X = x, \text{do}(A = a^\star)] \\
&= (\ell \circ g_0^{-1})(x) + \mathbb{E}[U|X = x, \text{do}(A = a^\star)] \\
&= (\ell \circ g_0^{-1})(x) + \mathbb{E}[U|g_0(Z) = x, \text{do}(A = a^\star)] \\
&= (\ell \circ g_0^{-1})(x) + \mathbb{E}[U|g_0(M_0 a^\star + V) = x, \text{do}(A = a^\star)] \\
&= (\ell \circ g_0^{-1})(x) + \mathbb{E}[U|V = g_0^{-1}(x) - M_0 a^\star, \text{do}(A = a^\star)] \\
&\overset{(*)}{=} (\ell \circ g_0^{-1})(x) + \mathbb{E}[U|V = g_0^{-1}(x) - M_0 a^\star] \\
&= ((\ell \circ \kappa_\phi^{-1}) \circ (\kappa_\phi \circ g_0^{-1}))(x) + \mathbb{E}[U|V = g_0^{-1}(x) - M_0 a^\star] \\
&\overset{(**)}{=} (\ell \circ \kappa_\phi^{-1})(\phi(x)) + \mathbb{E}[U|V = g_0^{-1}(x) - M_0 a^\star] \qquad (46)
\end{aligned}
$$

where the equality $(*)$ hold because $\forall a^\star \in \mathcal{A} : \mathbb{P}_{U,V} = \mathbb{P}_{U,V}^{\text{do}(A=a^\star)}$ and $(**)$ follows from the fact that $\phi$ aff-identifies $g_0^{-1}$. Next, define $h := v \mapsto H_\phi(v - \mathbb{E}[V])$. We have for all $x \in \text{Im}(g_0)$ and $a^\star \in \mathcal{A}$ that

$$
\begin{aligned}
h(g_0^{-1}(x) - M_0 a^\star) &= H_\phi(g_0^{-1}(x) - M_0 a^\star - \mathbb{E}[V]) \\
&= H_\phi g_0^{-1}(x) - H_\phi M_0 a^\star - H_\phi \mathbb{E}[V] \\
&= H_\phi g_0^{-1}(x) + c_\phi - (M_\phi a^\star + q_\phi) \\
&= (\phi \circ g_0 \circ g_0^{-1}(x)) - (M_\phi a^\star + q_\phi) \\
&= \phi(x) - (M_\phi a^\star + q_\phi) \\
&= \phi(x) - (W_\phi a^\star + \alpha_\phi). \qquad\qquad \text{from (39)} \qquad (47)
\end{aligned}
$$

Since the function $h$ is bijective, combining (47) and (46) yields

$$
\begin{aligned}
\mathbb{E}[Y|X = x, \text{do}(A = a^\star)] &= (\ell \circ \kappa_\phi^{-1})(\phi(x)) + \mathbb{E}[U|h(V) = h(g_0^{-1}(x) - M_0 a^\star)] \\
&= (\ell \circ \kappa_\phi^{-1})(\phi(x)) + \mathbb{E}[U|V_\phi = \phi(x) - (W_\phi a^\star + \alpha_\phi)] \\
&= (\ell \circ \kappa_\phi^{-1})(\phi(x)) + \lambda_\phi(\phi(x) - (W_\phi a^\star + \alpha_\phi)).
\end{aligned}
$$

Lastly, as argued in the first part of the proof, it holds from Theorem 2.3 in Newey et al. (1999) that $\nu \equiv (\ell \circ \kappa_\phi^{-1}) + \delta$ and $\psi \equiv \lambda_\phi - \delta$, for some constant $\delta \in \mathbb{R}$. We thus have that

$$
\forall x \in \text{Im}(g_0), a^\star \in \mathcal{A} : \mathbb{E}[Y|X = x, \text{do}(A = a^\star)] = \nu(\phi(x)) + \psi(\phi(x) - (W_\phi a^\star + \alpha_\phi)),
$$

which concludes the proof of the second statement. $\qquad\square$

### D.4 PROOF OF THEOREM 5

*Proof.* We begin the proof by showing the forward direction ($\phi$ satisfies (17) $\implies$ $\phi$ satisfies (6)). Let $\phi \in \Phi$ be an encoder that satisfies (17). We then have for all $a \in \text{supp}(A)$

$$
\begin{aligned}
W_\phi a + \alpha_\phi &= \mathbb{E}[\phi(X) \mid A = a] \\
&= \mathbb{E}[(\phi \circ g_0)(M_0 A + V) \mid A = a] \\
&= \mathbb{E}[(\phi \circ g_0)(M_0 a + V)] \qquad\qquad \text{since } A \perp\!\!\!\perp V.
\end{aligned}
$$

Define $h := \phi \circ g_0$. Taking derivative with respect to $a$ on both sides yields

$$
W_\phi = \frac{\partial \mathbb{E}[h(M_0 a + V)]}{\partial a}.
$$

Next, we interchange the expectation and derivative using the assumptions that $\phi$ and $g_0$ have bounded derivative and the dominated convergence theorem. We have for all $a \in \text{supp}(A)$

$$
\begin{aligned}
W_\phi &= \mathbb{E}\Big[\frac{\partial h(M_0 a + V)}{\partial a}\Big] \\
&= \mathbb{E}\Big[\frac{\partial h(u)}{\partial u}\Big|_{u = M_0 a + V} \frac{\partial(M_0 a + V)}{\partial a}\Big] \qquad\qquad \text{by the chain rule} \\
&= \mathbb{E}\Big[\frac{\partial h(u)}{\partial u}\Big|_{u = M_0 a + V} M_0\Big]. \qquad\qquad (48)
\end{aligned}
$$

Defining $h' : z \mapsto \left.\frac{\partial h(u)}{\partial u}\right|_{u=z}$ and $g : z \mapsto h'(z)M_0 - W_\phi$, we have for all $a \in \mathrm{supp}(A)$

$$0 = \mathbb{E}[h'(M_0 a + V)M_0 - W_\phi]$$
$$= \mathbb{E}[g(M_0 a + V)]$$
$$= \int g(M_0 a + v) f_V(v) d\, v.$$

Define $t := M_0 a \in \mathbb{R}^d$ and $\tau := t + v$, we then have for all $t \in \mathrm{supp}(M_0 A)$ that

$$0 = \int g(\tau) f_V(\tau - t) d\, (\tau - t)$$
$$= \int g(\tau) f_V(\tau - t) d\, \tau$$
$$= \int g(\tau) f_{-V}(t - \tau) d\, \tau. \tag{49}$$

Recall that $g$ is a function from $\mathbb{R}^d$ to $\mathbb{R}^{d \times k}$. Now, for an arbitrary $(i, j) \in \mathbb{R}^d \times \mathbb{R}^k$ define the function $g_{ij}(\cdot) : \mathbb{R}^d \to \mathbb{R} := g(\cdot)_{ij}$. We then have for each element $(i, j)$ and all $t \in \mathrm{supp}(M_0 A)$ that

$$0 = \int g_{ij}(\tau) f_{-V}(t - \tau) d\, \tau. \tag{50}$$

Next, let us define $c_{ij} : t \in \mathbb{R}^d \mapsto \int g_{ij}(\tau) f_{-V}(t - \tau) d\, \tau \in \mathbb{R}$. We now show that $c_{ij} \equiv 0$ where we adapt the proof of D'Haultfoeuille (2011, Proposition 2.3). By Assumption 2, $f_{-V}$ is analytic on $\mathbb{R}^d$, we thus have for all $\tau \in \mathbb{R}^d$ that the function $t \mapsto g_{ij}(\tau) f_{-V}(t - \tau)$ is analytic on $\mathbb{R}^d$. Moreover, since $g_{ij}$ is bounded the function $t \mapsto g_{ij}(\tau) f_{-V}(t - \tau)$ is bounded, too. Thus, by (Rudin, 1987, page 229), the function $c_{ij}$ is then also analytic on $\mathbb{R}^d$. Using that $M_0$ is surjective, we have by the open mapping theorem (see e.g., Bühler & Salamon (2018), page 54) that $M_0$ is an open map. Now, since $\mathrm{supp}(A)$ contains a non-empty open subset of $\mathbb{R}^k$ and $M_0$ is an open map, we thus have from (50) that $c_{ij}(t) = 0$ on a non-empty open subset of $\mathbb{R}^d$. Then, by the identity theorem, the function $c_{ij}$ is identically zero, that is,

$$c_{ij} \equiv 0. \tag{51}$$

Next, we show that $g_{ij} \equiv 0$. Let $L^1$ denote the space of equivalence classes of integrable functions from $\mathbb{R}^d$ to $\mathbb{R}$. For all $t \in \mathbb{R}^d$, let us define $f_t(\cdot) := f_{-V}(t - \cdot)$ and $Q := \{f_t \mid t \in \mathbb{R}^d\}$. By Assumption 2, the characteristic function of $V$ does not vanish. This implies that the characteristic function of $-V$ does not vanish either (since the characteristic function of $-V$ is the complex conjugate of the characteristic function of $V$). We therefore have that the Fourier transform of $f_{-V}$ has no real zeros. Then, we apply Wiener's Tauberian theorem (Wiener, 1932) and have that $Q$ is dense in $L^1$. Using that $Q$ is dense in $L^1$, combining with (51) and the continuity of the linear form $\tilde{\phi} \in L^1 \mapsto \int g_{ij}(\tau) \tilde{\phi}(\tau) d\, \tau$ (continuity follows from boundedness of $g_{ij}$ and Cauchy-Schwarz), it holds that

$$\forall \tilde{\phi} \in L^1 : \int g_{ij}(\tau) \tilde{\phi}(\tau) d\, \tau = 0. \tag{52}$$

From (52), we can then conclude that

$$g_{ij}(\cdot) \equiv 0. \tag{53}$$

Next, from (53) and the definition of $g$, we thus have for all $a \in \mathrm{supp}(A)$ and $v \in \mathbb{R}^d$

$$h'(M_0 a + v)M_0 = W_\phi. \tag{54}$$

As $M_0$ has full row rank, it thus holds that

$$h'(M_0 a + v) = W_\phi M_0^\dagger. \tag{55}$$

We therefore have that the function $h = \phi \circ g_0$ is an affine transformation. Furthermore, using that $g_0$ is injective and $\phi|_{\mathrm{Im}(g_0)}$ is bijective, the composition $h = \phi \circ g_0$ is also injective. Therefore, there exists an invertible matrix $H \in \mathbb{R}^{d \times d}$ and a vector $c \in \mathbb{R}^d$ such that

$$\forall z \in \mathbb{R}^d : \phi \circ g_0(z) = Hz + c, \tag{56}$$

which concludes the proof of the forward direction.

Next, we show the backward direction of the statement ($\phi$ satisfies (6) $\implies$ $\phi$ satisfies (17)). Let $\phi \in \Phi$ satisfy (17). Then, there exists an invertible matrix $H \in \mathbb{R}^{d \times d}$ and a vector $c \in \mathbb{R}^d$ such that $\forall z \in \mathbb{R}^d : (\phi \circ g_0)(z) = Hz + c$. We first show the second condition of (17). By the invertibility of $H$, the composition $\phi \circ g_0$ is bijective. By Lemma 10, we thus have that $\phi|_{\mathrm{Im}(g_0)}$ is bijective. Next, we show the first condition of (17). Let $\mu_V := \mathbb{E}[V]$. We have for all $\alpha \in \mathbb{R}^d, W \in \mathbb{R}^{d \times d}$

$$
\begin{aligned}
\mathbb{E}[&\|\phi(X) - \alpha - WA\|^2] \\
&= \mathbb{E}[\|(\phi \circ g_0)(Z) - \alpha - WA\|^2] \\
&= \mathbb{E}[\|HZ + c - \alpha - WA\|^2] \\
&= \mathbb{E}[\|H(M_0 A + V) + c - \alpha - WA\|^2] \\
&= \mathbb{E}[\|(HM_0 - W)A + (c - \alpha) + HV\|^2] \\
&= \mathbb{E}[\|(HM_0 - W)A + (c + H\mu_V - \alpha) + H(V - \mu_V)\|^2] \\
&= \mathbb{E}[\|(HM_0 - W)A + (c + H\mu_V - \alpha)\|^2] \\
&\qquad + 2\,\mathbb{E}[((HM_0 - W)A + (c + H\mu_V - \alpha))^\top H(V - \mu_V)] + \mathbb{E}[\|H(V - \mu_V)\|^2] \\
&= \mathbb{E}[\|(HM_0 - W)A + (c + H\mu_V - \alpha)\|^2] + \mathbb{E}[\|H(V - \mu_V)\|^2]. \qquad \text{since } A \perp\!\!\!\perp V
\end{aligned}
$$

Since the covariance matrix of $A$ is full rank, we therefore have that

$$
(\alpha_\phi, W_\phi) \overset{def}{=} \underset{\alpha \in \mathbb{R}^d, W \in \mathbb{R}^{d \times k}}{\mathrm{argmin}} \mathbb{E}[\|\phi(X) - \alpha - WA\|^2] = (c + H\mu_V, HM_0). \tag{57}
$$

Then, we have for all $a \in \mathrm{supp}(A)$ that

$$
\begin{aligned}
\mathbb{E}[\phi(X) - \alpha_\phi - W_\phi A \mid A = a] &\overset{(*)}{=} \mathbb{E}[(\phi \circ g_0)(Z) - (c + H\mu_V) - HM_0 A \mid A = a] \\
&= \mathbb{E}[HZ + c - (c + H\mu_V) - HM_0 A \mid A = a] \\
&= \mathbb{E}[H(M_0 A + V) + c - (c + H\mu_V) - HM_0 A \mid A = a] \\
&= \mathbb{E}[HV - H\mu_V \mid A = a] \\
&\overset{(**)}{=} H\mu_V - H\mu_V \\
&= 0,
\end{aligned}
$$

where the equality $(*)$ follows from (57) and $(**)$ holds by $A \perp\!\!\!\perp V$. This concludes the proof. $\qquad\square$

# E  DETAILS ON THE ALGORITHM

## E.1  THE ALGORITHM FOR REP4EX

We here present a pseudo algorithm for `Rep4Ex-CF`, see Algorithm 1.

## E.2  HEURISTIC FOR CHOOSING REGULARIZATION PARAMETER $\lambda$

To select the regularization parameter $\lambda$ in the regularized auto-encoder objective function (21), we employ the following heuristic. Let $\Lambda = \{\lambda_1, \ldots, \lambda_m\}$ be our candidate regularization parameters, ordered such that $\lambda_1 > \lambda_2 > \cdots > \lambda_m$. For each $\lambda_i$, we estimate the minimizer of (21) and calculate the reconstruction loss. Additionally, we compute the reconstruction loss when setting $\lambda = 0$ as the baseline loss. We denote the resulting reconstruction losses for different $\lambda_i$ as $R_{\lambda_i}$ (and $R_0$ for the baseline loss). Algorithm 2 illustrates how $\lambda$ is chosen.

In our experiments, we set a cutoff parameter at 0.2 and for each setting execute the heuristic algorithm only during the first repetition run to save computation time. Figure 4 demonstrates the effectiveness of our heuristic. Here, our algorithm would suggest choosing $\lambda = 10^2$, which also corresponds to the highest R-squared value.

Another approach to choose $\lambda$ is to apply the conditional moment test in Muandet et al. (2020) to test whether the linear invariance constraint (16) is satisfied. Specifically, in a similar vein to Jakobsen

---

**Algorithm 1:** An algorithm for Rep4Ex

---

**Input:** observations $(x_i, a_i, y_i)_{i=1}^n$, target interventions $(a_j^\star)_{j=1}^m$, auto-encoder AE, additive
       regression AR

// Train the auto-encoder

$\phi = \text{AE}((x_i, a_i)_{i=1}^n)$ ;

// Regress $\phi(X)$ on $A$

$(\hat{W}_\phi, \hat{\alpha}_\phi) = \text{argmin}_{W,\alpha} \sum_{i=1}^n \|\phi(x_i) - (Wa_i + \alpha)\|^2$ ;

// Obtain the control variables

**for** $i = 1$ **to** $n$ **do**
   |  $v_i = \phi(x_i) - (Wa_i + \alpha)$ ;
**end**

// Train additive regression

$\hat{\nu}, \hat{\psi} = \text{AR}(y_i \sim \nu(\phi(x_i)) + \psi(v_i), i = 1 \dots n)$ ;

// Estimate $\mathbb{E}[Y | \text{do}(A = a^\star)]$

**for** $j = 1$ **to** $m$ **do**
   |  $\hat{y}_j = \sum_{i=1}^n \hat{\nu}(\hat{W}_\phi a_j^\star + \hat{\alpha}_\phi + v_i) - \sum_{i=1}^n \left( \hat{\nu}(\phi(x_i)) - y_i \right)$
**end**

**Output:** $(\hat{y}_j)_{j=1}^m$

---

**Algorithm 2:** Choosing $\lambda$ parameter

---

**Input:** cut off parameter $\alpha$

$\lambda \leftarrow \lambda_m$ ;

**for** $i = 1$ **to** $m - 1$ **do**
   |  $\delta_i = \frac{R_{\lambda_i}}{R_0} - 1$ ;
   |  **if** $\delta_i < \alpha$ **then**
   |    |  $\lambda \leftarrow \lambda_i$ ;
   |    |  **break**
**return** $\lambda$

---

& Peters (2022); Saengkyongam et al. (2022), we may select the smallest possible value of $\lambda$ for which the conditional moment test is not rejected.

## F    POSSIBLE WAYS OF CHECKING APPLICABILITY OF THE PROPOSED METHOD

Due to the nature of extrapolation problems, it is not feasible to definitively verify the method's underlying assumptions from the training data. However, we may still be able to check and potentially falsify the applicability of our approach in practice. To this end, we propose comparing its performance under two different cross-validation schemes:

  (i) Standard cross-validation, where the data is randomly divided into training and test sets.

  (ii) Extrapolation-aware cross-validation, in which the data is split such that the support of $A$ in the test set does not overlap with that in the training set.

By comparing our method's performance across these two schemes, we can assess the applicability of our overall method. A significant performance gap may suggest that some key assumptions are not valid and one could consider adapting the setting, e.g., by transforming $A$ (see Remark 6).

A further option of checking for potential model violations is to test for linear invariance of the fitted encoder, using for example the conditional moment test by Muandet et al. (2020). If the null hypothesis of linear invariance is rejected, this indicates that either the optimization was unsuccessful or the model is incorrectly specified.

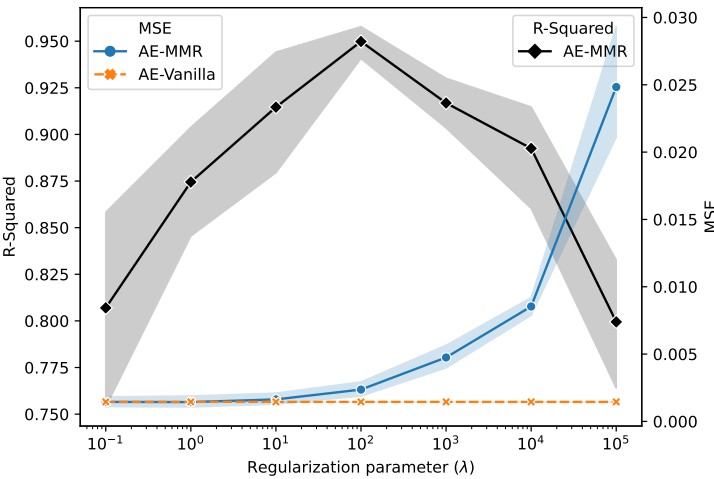

Figure 4

## G DETAILS ON THE EXPERIMENTS

### G.1 DATA GENERATING PROCESSES (DGPs) IN SECTION 6

**DGP for Section 6.1**  We consider the following underlying SCM

$$\mathcal{S}(\alpha): \quad \left\{ A := \epsilon_A \qquad Z := \alpha M_0 A + V \qquad X := g_0(Z) \right. \tag{58}$$

where $\epsilon_A \sim \text{Unif}(-1, 1)$ and $V \sim N(0, \Sigma)$ are independent noise variables. Here, we consider a four-layer neural networks with Leaky ReLU activation functions as the mixing function $g_0$. The parameters of the neural networks and the parameters of the SCM (22) including $\Sigma$ and $M_0$ are randomly chosen, see below for more details. The parameter $\alpha$ controls the strength of the effect of $A$ on $Z$. In this experiment, we set the dimension of $X$ to 10 and consider two choices $d \in \{2, 4\}$ for the dimension of $Z$. Additionally, we set the dimension of $A$ to the dimension of $Z$.

**DGP for Section 6.2**  We consider the following underlying SCM

$$\mathcal{S}(\gamma): \quad \left\{ A := \epsilon_A^\gamma \qquad Z := M_0 A + V \qquad X := g_0(Z) \qquad Y := \ell(Z) + U. \right. \tag{59}$$

where $\epsilon_A^\gamma \sim \text{Unif}([-\gamma, \gamma]^k)$ and $V \sim N(0, \Sigma_V)$ are independent noise variables. $U$ is then generated as $U := h(V) + \epsilon_U$, where $\epsilon_U \sim N(0, 1)$. The parameter $\gamma$ determines the support of $A$ in the observational distribution. Similar to Section 6.1, we consider a four-layer neural networks with Leaky ReLU activation functions as the mixing function $g_0$ and the parameters of $g_0$, $\Sigma_V$, and $M_0$ are randomly chosen as detailed below.

**Details on other parameters**  In all experiments, we employ a neural network with the following details as the mixing function $g_0$:

- Activation functions: Leaky ReLU
- Architecture: three hidden layers with the hidden size of 16
- Initialization: weights are independently drawn from $\text{Unif}(-1, 1)$.

As for the matrix $M_0$, each element is indepedently drawn from $\text{Unif}(-2, 2)$. The covariance $\Sigma_V$ is generated by $\Sigma_V := AA^\top + \text{diag}(V)$, where $A$ and $V$ are indepedently drawn from $\text{Unif}([0, 1]^d)$.

In Section 6.2, in the case of one-dimensional $A$, we specify the functions $h$ and $\ell$ as $h : v \mapsto \frac{1}{5}v^3$, $\ell : z \mapsto -2z + 10\sin(z)$. In the case of multi-dimensional $A$ (see Appendix H.1), we employ the following neural network for both $h$ and $\ell$:

- Activation functions: Tanh

- Architecture: one hidden layer with the hidden size of 64
- Initialization: weights are independently drawn from $\text{Unif}(-1, 1)$.

Lastly, in all experiments, we use the Gaussian kernel for the MMR term in the objective function (21). The bandwidth of the Gaussian kernel is chosen by the median heuristic (e.g., Fukumizu et al., 2009).

## G.2 AUTO-ENCODER DETAILS

We employ the following hyperparameters for all autoencoders in our experiments. The same architecture is utilized for both the encoder and decoder:

- Activation functions: Leaky ReLU
- Architecture: three hidden layers with the hidden size of 32
- Learning rate: 0.005
- Batch size: 256
- Optimizer: Adam optimizer with $\beta_1 = 0.9, \beta_2 = 0.999$
- Number of epochs: 1000.

For the variational auto-encoder, we employ a standard Gaussian prior with the same network architecture and hyperparameters as defined above.

## H FURTHER DETAILS ON EXPERIMENTAL RESULTS

Figures 5 and 6 show reconstruction performance of the hidden variables for the experiment described in Section 6.1.

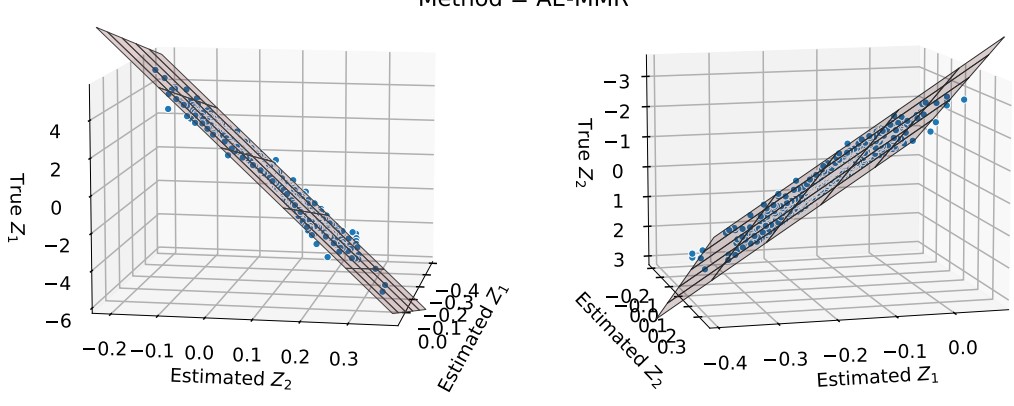

Figure 5

## H.1 SECTION 6.2 CONTINUED: MULTI-DIMENSIONAL $A$

Here, we consider multi-dimensional variables $Z$ and $A$. Similar to Section 6.1, we set the dimension of $X$ to 10, vary the dimension $d$ of $Z$, and keep the dimension of $A$ equal to that of $Z$. We specify the functions $h$ and $\ell$ using two-layer neural networks with the hyperbolic tangent activation functions. For the training distribution, we generate $A$ from a uniform distribution over $[-1, 1]^d$. To assess extrapolation performance, we generate 100 test points of $A$ from a uniform distribution over $[-3, -1]^d$ and calculate the mean squared error with respect to the true conditional mean. In addition to the baseline MLP, we also include an oracle method, denoted as Rep4Ex-CF-Oracle, where we directly use the true latent predictors $Z$ instead of learning a representation in the first

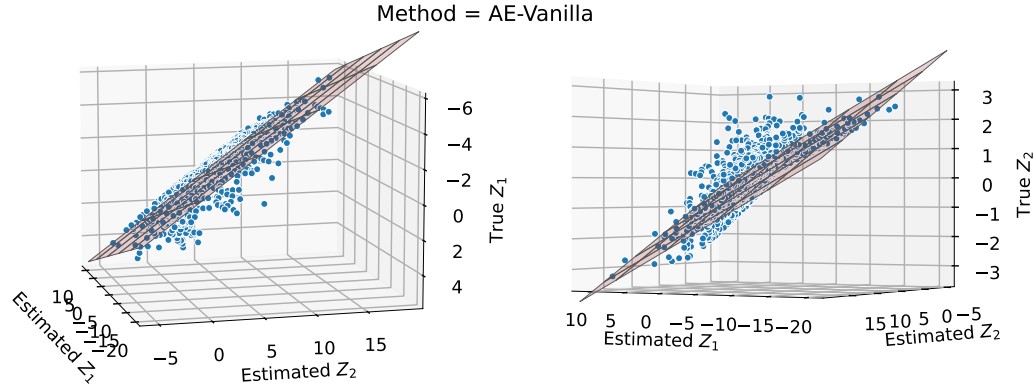

Figure 6

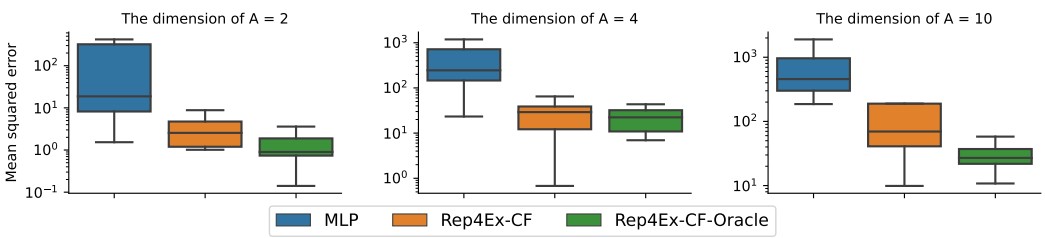

Figure 7: MSEs of different methods for three distinct dimensionalities of $A$. The box plots illustrate the distribution of MSEs based on 10 repetitions. `Rep4Ex-CF` yields substantially lower MSEs in comparison to the baseline `MLP`. Furthermore, the MSEs achieved by `Rep4Ex-CF` are comparable to those of `Rep4Ex-CF-Oracle`, underscoring the effectiveness of the representation learning stage.

stage. The outcomes for $d \in \{2, 4, 10\}$ are depicted in Figure 7. Across all settings, our proposed method, `Rep4Ex-CF`, consistently achieves markedly lower mean squared error compared to the baseline `MLP`. Furthermore, the performance of `Rep4Ex-CF` is on par with that of the oracle method `Rep4Ex-CF-Oracle`, indicating that the learned representations are close to the true latent predictors (up to an affine transformation).

## H.2 SECTION 6.2 CONTINUED: IMPACT OF UNOBSERVED CONFOUNDERS

Our approach allows for unobserved confounders between $Z$ and $Y$. This section explores the impact of such confounders on extrapolation performance empirically. We consider the SCM as in (59) from Section 6.2, where we set $\gamma = 1.2$ and generate the noise variables $U$ and $V$ from a joint Gaussian distribution with the covariance matrix $\Sigma_{U,V} = \begin{pmatrix} 1 & \rho \\ \rho & 1 \end{pmatrix}$. Here, the parameter $\rho$ controls the dependency between $U$ and $V$, representing the strength of unobserved confounders. Figure 8 presents the results for four different confounding levels $\rho = (0, 0.1, 0.5, 0.9)$. Our method, `Rep4Ex-CF`, demonstrates robust extrapolation capabilities across all confounding levels.

## H.3 SECTION 6.2 CONTINUED: ROBUSTNESS AGAINST VIOLATING THE MODEL ASSUMPTION OF NOISELESS $X$

In Setting 1, we assume that the observed features $X$ are deterministically generated from $Z$ via the mixing function $g_0$. However, this assumption may not hold in practice. In this section, we investigate the robustness of our method against the violation of this assumption. We conduct an experiment with the setting similar to that with one-dimensional $A$ in Section 6.2 but here we introduce independent additive random standard Gaussian noise in $X$, i.e., $X := g_0(Z) + \epsilon_X$, where $\epsilon_X \sim N(0, \sigma^2 I_m)$. The parameter $\sigma$ controls the noise level. Figure 9 illustrates the results for

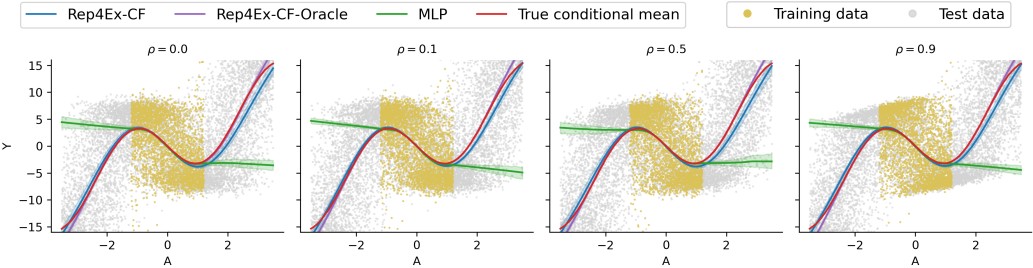

Figure 8: Different estimations of the target of inference $\mathbb{E}[Y \mid \text{do}(A := \cdot)]$ as the strength of unobserved confounders ($\rho$) increases. Notably, the extrapolation performance of `Rep4Ex-CF` remains consistent across all confounding levels.

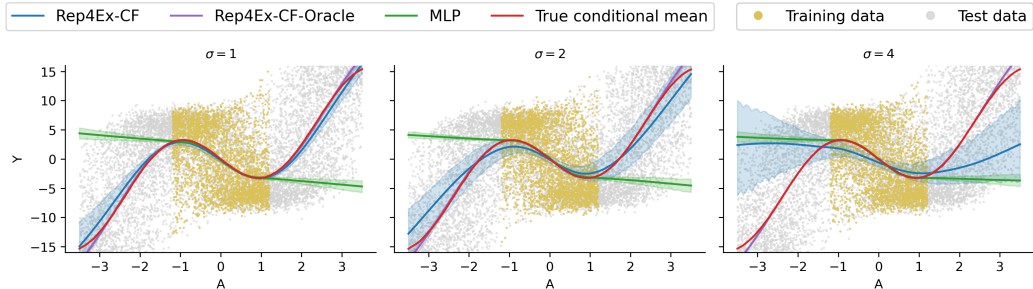

Figure 9: Different estimations of the target of inference $\mathbb{E}[Y \mid \text{do}(A := \cdot)]$ in the presence of noise in $X$. Our method, `Rep4Ex-CF`, demonstrates the ability to extrapolate beyond the training support when the noise is not too large, suggesting the potential to relax the assumption of the absence of noise in $X$.

different noise levels $\sigma = (1, 2, 4)$. The results indicate that our method maintains successful extrapolation capabilities under moderate noise conditions. Therefore, we believe it may be possible to relax the assumption of the absence of noise in $X$.

