# OpenReview forum: "Identifying Representations for Intervention Extrapolation"
_ICLR.cc/2024/Conference — ICLR 2024 poster_

### Official Review · Reviewer_wfVE · 2023-10-20

**Soundness:** 4 excellent
**Presentation:** 4 excellent
**Contribution:** 4 excellent
**Rating:** 8
**Confidence:** 4

**Summary:**

This work introduces a type of interventional extrapolation which consists in predicting the effect of an unseen intervention. The causal model considered is given by A -> Z -> X and Z->Y where Z is latent, A -> Z is linear, Z->X is invertible and Z -> Y is an additive noise model. The goal is then to estimate E[Y | do(A = a*)] where a* is outside its training domain. The authors separate this problem in three subproblems: 1) estimation given that Z is observed, 2) estimation this given that Z is known only up to an affine transformation, and 3) how to identify the invertible map Z->X up to an affine transformation. These identifiability results are then transferred into a practical algorithm: a) The mixing function is recovered by performing SGD on a loss consisting of a reconstruction term and a regularizer based on maximum moment restriction (MMR), and b) the learned encoder is then used to estimate the desired “do” expectation following the procedure of step 2) above.  The proposed estimation procedure is then validated on a synthetic dataset.

Review Summary: I gave the main text a thorough reading and checked the large majority of the math it presents. Despite the few problems I raised in my review, I believe this work is sufficiently novel, interesting, rigorous and well written to warrant acceptance to ICLR. I was planning to give a score of 7, since this score is not available, I'm rounding up to 8. I'm giving this score assuming that a discussion of the limitations will be added, as I suggest in my review.

**Strengths:**

- I agree that we need more theory to demonstrate concrete benefits of identifiable representation learning. I enjoyed this perspective.
- The paper is very clearly written and structured and relatively easy to follow despite its technicality. The level of mathematical rigor in this work is very high in my opinion and the notation is always very crisp and transparent.
- I thought the theoretical contributions were novel and interesting.
- I appreciated that the main ideas about the proofs were provided in the main text. Most works in this literature only give very short proof sketches if any.
- Theoretical results are well modularized to facilitate reuse.
- I thought the method to learn the encoder up to affine transformation was novel and interesting.
- I’m left with a good feeling of having learned something new.

**Weaknesses:**

- The limitations of the proposed approach could be discussed further. For example, how does the method perform under various assumption violations? A few experiments could provide some insight into this.
- The paper ends a bit abruptly. There’s no conclusion nor discussion of limitations. I guess this is the cost of having so much technical details in the main text (which, as I said, I appreciated, but I’m unsure whether this is a good balance.)
- I think the paper could do a better job of contrasting its results with what already appears in the literature, especially regarding the affine identifiability results. How does this compare to iVAE for example, which has a very similar data generating process with an auxiliary variable (which would be A here)? A few papers have similar results: [1,2,3,4].
- I couldn’t grasp a few reasoning steps in the main text (see Questions below).

Minor:
- $M_0$ has full row rank implies that dim(A) >=  dim(Z). Might be worth making explicit.
- I always forget which noise variable U or V is associated with which variable. How about replacing V by V_z and U by V_y?
- Footnote one: what is $\mathcal{B}$?
- The introduction rightfully points out that the literature has almost no works giving theoretical arguments for why the identifiable representation learning is important. You might want to consider citing [1, 5] as examples of work pushing that direction.

[1] S. Lachapelle, T. Deleu, D. Mahajan, I. Mitliagkas, Y. Bengio, S. Lacoste-Julien, and Q. Bertrand. Synergies between disentanglement and sparsity: a multi-task learning perspective, 2022.

[2] I. Khemakhem, D. Kingma, R. Monti, and A. Hyvarinen. Variational autoencoders and nonlinear ¨ ica: A unifying framework. In Proceedings of the Twenty Third International Conference on Artificial Intelligence and Statistics, 2020.

[3] Ahuja, K., Mahajan, D., Syrgkanis, V., and Mitliagkas, I. Towards efficient representation identification in supervised learning. In First Conference on Causal Learning and Reasoning, 2022

[4] Roeder, G., Metz, L., and Kingma, D. P. On linear identifiability of learned representations. In Proceedings of the 38th International Conference on Machine Learning, 2021.

[5] S. Lachapelle, D. Mahajan, I. Mitliagkas, and S. Lacoste-Julien. Additive decoders for latent variables identification and cartesian-product extrapolation. In Advances in Neural Information Processing Systems, 2023b.

**Questions:**

- Lemma 7 in Appendix A: Could you explain the argument a bit more? I’m unaware of this proof technique, so citing the result used here would be useful. Or is it an alternative definition of conditional independence that I’m not aware of?
- Paragraph after (3):
    - Could you give some insight as to why $\ell$ and $\lambda$ are not identifiable up to additive constant without further assumptions in (3)? I’m not sure I see what could go wrong since we observe Y, Z and V here (V is identifiable). Also, can you state the assumption from Newey et al. (1999) that allows you to identify $\ell$? I suspect it has something to do with the differentiability + supp(A) convex assumption from Theorem 4?
    - It is written “for all $a^* \in M_0 supp(A) + supp(V)$” somewhere, but it seems wrong since $M_0 supp(A) + supp(V)$ is an event for Z, no? I’m not sure I’m following this part of the argument and onward.
    - Same paragraph: I think we need to compute $\ell(z)$ only for $z \in M_0\mathcal{A} + supp(V)$, no?
- Eq. (16): I’m not sure why this holds. Is it because the inside of the conditional expectation is equal to V? But we don’t assume E[V] = 0 right? What am I missing? Since this is so crucial to the algorithm, it might be worth expliciting further.

---

> ### Author Response · Authors · 2023-11-20
> **Response to Reviewer wfVE**
>
> We thank the reviewer for the positive and detailed feedback.
>
> We address the points raised in the 'Weaknesses' and 'Questions' sections below.
>
> 1. > The limitations of the proposed approach could be discussed further. For example, how does the method perform under various assumption violations? A few experiments could provide some insight into this.
>
>     We have now added Remark 6 in Appendix B which highlights the key assumptions from Setting 1 and discusses whether they can be extended (and have referenced this appendix after Setting 1). Additionally, we have conducted an additional experiment in which we introduced random noise into $X$. The results indicate that our method is robust to the presence of noise in $X$ (when the noise is not too large). The results can now be found in Appendix H.3. We have also noted in the conclusion that future work could extend our theoretical results to cover this case.
>
> 2. > The paper ends a bit abruptly. There’s no conclusion nor discussion of limitations. I guess this is the cost of having so much technical details in the main text (which, as I said, I appreciated, but I’m unsure whether this is a good balance.)
>
>     Thank you for your suggestion. We have now included a discussion section that summarizes the main focuses of our work and discusses potential extensions of our theoretical results.
>
> 3. > I think the paper could do a better job of contrasting its results with what already appears in the literature, especially regarding the affine identifiability results. How does this compare to iVAE for example, which has a very similar data generating process with an auxiliary variable (which would be A here)? A few papers have similar results: [1,2,3,4].
>
>     We agree that our actions $A$ could be considered auxiliary variables. However, the work by Khemakhem et al., 2020 differs from ours in two important ways.
>     * Assumptions: one of the key assumptions in their setting is that the density $p(z|a)$ is conditionally factorized, meaning that the components of $Z$ are independent when conditioned on $A$. In contrast, our approach permits dependence among the components of $Z$ even when conditioned on $A$ (because in our setting, the components of $V$ can have arbitrary dependencies).
>     * Type of identifiability: maybe even more importantly, Khemakhem et al. provide identifiability up to point-wise nonlinearities which is not sufficient for intervention extrapolation. The main focus of our work is to provide an identification that facilitates a solution to the task of intervention extrapolation.
>
>     We have now included a discussion on the relationship to the nonlinear ICA literature in Appendix A and have referenced this appendix in the related work section.
>
> 4. > $M_0$ has full row rank implies that dim(A) >= dim(Z). Might be worth making explicit.
>
>     Thank you for the suggestion. We have now added this comment in Setting 1.
>
> 5. > I always forget which noise variable U or V is associated with which variable. How about replacing V by V_z and U by V_y?
>
>     We have decided to retain our current notation, as introducing a subscript might lead to additional notational complexities (for instance, we already use $V_{\theta}$). We recall our notation using “'U' precedes 'V' and 'Y' precedes 'Z'”.
>
> 6. > Footnote one: what is $\mathcal{B}$
>
>     We have now clarified Footnote 1 and replaced the letter $\mathcal{B}$ by $\Omega$ since $\mathcal{B}$ has already been used in Proposition 1.
>
> 7. > The introduction rightfully points out that the literature has almost no works giving theoretical arguments for why the identifiable representation learning is important. You might want to consider citing [1, 5] as examples of work pushing that direction.
>
>     Thank you for the pointer to these references; we have now included them in the introduction.
> 8. > Lemma 7 in Appendix A: Could you explain the argument a bit more? I’m unaware of this proof technique, so citing the result used here would be useful. Or is it an alternative definition of conditional independence that I’m not aware of?
>
>     This is an alternative way of defining conditional independence (in the absence of densities). We have now clarified the proof and have added a citation to a paper by Constantinou & Dawid, 2017 that proves the corresponding equivalence – which allowed us to simplify our proof.

---

> ### Author Response · Authors · 2023-11-20
> **Response to Reviewer wfVE (continued)**
>
> 9. > Could you give some insight as to why $\ell$ and $\lambda$ are not identifiable up to additive constant without further assumptions in (3)? I’m not sure I see what could go wrong since we observe Y, Z and V here (V is identifiable). Also, can you state the assumption from Newey et al. (1999) that allows you to identify $\ell$? I suspect it has something to do with the differentiability + supp(A) convex assumption from Theorem 4?
>
>    A simple counter example is when $\ell$ and $\lambda$ are linear functions, i.e., $\mathbb{E}[Y | Z, V] = \beta_1 Z + \beta_2 V$ and $Z$ and $V$ have a perfect linear relationship. This scenario leads to perfect multicollinearity in linear regression, which results in the non-identifiability of $\beta_1$ and $\beta_2$. We therefore require additional assumptions that $\ell$ and $\lambda$ are differentiable and the interior of $\text{supp}(A)$ is convex.
>
> 10. > It is written “for all $a^* \in M_0supp(A) + supp(V)$" somewhere, but it seems wrong ...
>
>     Many thanks for spotting this typo. We have now corrected it and write for all $a^*$ such that $M_0a^* + \text{supp}(V) \subseteq M_0\text{supp}(A) + \text{supp}(V)$.
>
> 11. > Same paragraph: I think we need to compute $\ell(z)$ only for $z \in M_0 \mathcal{A} + supp(V)$
>
>     You are right, it is more accurate to write $z \in M_0 \mathcal{A} + \text{supp}(V)$. We have now simplified and clarified this paragraph.
>
> 12. > Eq. (16): I’m not sure why this holds. Is it because the inside of the conditional expectation is equal to V? But we don’t assume E[V] = 0 right? What am I missing? Since this is so crucial to the algorithm, it might be worth expliciting further.
>
>     The residual has zero mean since we have an intercept term $\alpha_{g^{-1}_0}$ that cancels out the mean of $V$.

---

> > ### Comment · Reviewer_wfVE · 2023-11-22
> >
> > Thanks for your answer.

---

### Official Review · Reviewer_dGv9 · 2023-10-31

**Soundness:** 3 good
**Presentation:** 4 excellent
**Contribution:** 3 good
**Rating:** 8
**Confidence:** 4

**Summary:**

The paper discusses identification strategies for effect extrapolation when the treatment of interest is unobserved during the experiment. The key assumptions are that

- (exogenous) treatment A affects outcome Y through and only through an unobserved mediator Z
- can observe a rich feature $X=g_0(Z)$ that is and only is a (injective) function of Z
- the relationship between A and Z is linear

Together with some other regularity assumptions, the author shows that, given an encoder that aff-identifies $g_0^{-1}$, it is possible to identify $E[Y|do(A=a^*)]$ and $E[Y|X=x,do(A=a^*)]$ for a treatment $a^*$ that has never been observed. They also propose a method for locating such an encoder.

**Strengths:**

- The paper studies intervention extrapolation under a well-chosen set of assumptions, which I find more appealing than the previously studied scenarios in the literature.

- To the best of my knowledge, the proposed identification strategy is novel.

- The manuscript is clear and well-written.

- The proposed algorithm is straightforward and practical.

**Weaknesses:**

- The assumptions are clear mathematically but might seem opaque to readers unfamiliar with the literature. The authors may want to give an example of what some of the key assumptions would imply in a simple setup.

- Many of the structural assumptions are not testable, and it is unclear to me when one shall be comfortable using the proposed method.

Also see questions.

**Questions:**

- It seems that this approach relies heavily on the linear structure between A and Z. Can this structural model be extended to $M_0 t(A) + V$ for some transformation $t(A)$ (e.g., $A^2$)? What if the relationship between A and Z is not linear, but rather through a known class of models $m_{\theta}(A)+V$ with parameters $\theta$? How would this affect the result?

- Is it possible to view the linear assumption as an approximation through a Taylor expansion around supp(A)? How would this compare with an extrapolation that is solely based on Lipschitz assumptions (see, e.g., Ben-Michael et al. 2021)?

- Although this paper is only about identification, I am curious about how the errors would accumulate.

- I find the sudden change of notation in Theorem 4 confusing.

- Can there be randomness in X, i.e., can X be a noisy observation of $g_0(Z)$?

- On page 2, the authors wrote "we allow for potentially unobserved confounders between Y and Z". How could there be confounding when the action is exogenous?

- Although the goal is very different, the approach reminds me of the negative control literature. The authors may want to discuss the connection.

---

> ### Author Response · Authors · 2023-11-20
> **Response to Reviewer dGv9**
>
> We thank the reviewer for the positive feedback.
>
> We address the points raised in the 'Weaknesses' and 'Questions' sections below.
>
> 1. > The assumptions are clear mathematically but might seem opaque to readers unfamiliar with the literature. The authors may want to give an example of what some of the key assumptions would imply in a simple setup.
>
>     Thanks for the suggestion. We have now added Remark 6 in Appendix B which highlights the key assumptions from Setting 1 and discusses whether they can be extended.
>
> 2. > Many of the structural assumptions are not testable, and it is unclear to me when one shall be comfortable using the proposed method.
>
>     Due to the nature of extrapolation problems, it is not feasible to definitively verify the method's underlying assumptions from the training data. However, we may still be able to check and potentially falsify the applicability of our approach in practice. To this end, we propose comparing its performance under two different cross-validation schemes:
>     * Standard cross-validation, where the data is randomly divided into training and test sets.
>     * Extrapolation-aware cross-validation, in which the data is split such that the support of $A$ in the test set does not overlap with that in the training set.
>
>     By comparing our method's performance across these two schemes, we can assess the applicability of our overall method. If the standard cross-validation yields a much better score, this may suggest that some key assumptions are not valid and one should consider adapting the setting, e.g., by transforming $A$ (see Remark 6 in Appendix B).
>
>     ##
>     A further option of checking for potential model violations is to test for linear invariance of the fitted encoder, using for example the conditional moment test by Muandet et al. (2020). If the null hypothesis of linear invariance is rejected, this indicates that either the optimization was unsuccessful or the model is incorrectly specified.
>
>     ##
>     We have now added the above discussion in Appendix F.
>
> 3. > It seems that this approach relies heavily on the linear structure between A and Z. Can this structural model be extended to ...
>
>     We have now added Remark 6 in Appendix B (and have referenced this appendix after Setting 1) to discuss key assumptions in Setting 1 and whether they can be extended. Some extensions are straightforward; for example, one can relax the linearity assumption from $A$ to $Z$ if there is a known nonlinear transformation $h$ such that $Z := M_0 h(\tilde A) + V$; in this case, we use $A := h(\tilde A)$ and obtain a model that lies in our model class.  Furthermore, our setting accommodates any invertible nonlinear transformation on $Z$, too. To see this, if there is a nonlinear injective function $h$ such that $\tilde{Z} := h(M_0 A + V)$ and $X := g_0(\tilde{Z})$, we can define $Z := M_0 A + V$ and $X := (g_0 \circ h)(Z)$ and obtain a model in our model class. But key to our approach is that the model of $Z$ on $A$ has the ability to extrapolate, otherwise non-linear extrapolation for $\mathbb{E}[Y|\text{do}(A:=a^*)]$ is not possible.
>
> 4. > Is it possible to view the linear assumption as an approximation through a Taylor expansion around supp(A)? How would this compare with an extrapolation that is solely based on Lipschitz assumptions (see, e.g., Ben-Michael et al. 2021)?
>
>     In our work, the extrapolation stems from the fact that the support of $V$ expands the support of $Z$ beyond $M_0\text{supp}(A)$. Together with the linearity assumption from $Z$ on $A$ (which can be composed with a known nonlinearity, as discussed in Remark 6 in Appendix B), one can then decompose the effects from $V$ and $A$ and hence extrapolate. In our view, this type of extrapolation is different from extrapolation based on Taylor expansion or smoothness constraints, as these approaches use regularity of the nonlinear functions, while our approach relies on the support extension of $V$ to make the nonlinearity identifiable. We have now added a comment on this right before Section 3.
>
> 5. > Although this paper is only about identification, I am curious about how the errors would accumulate.
>
>     Appendix E.1 of the original submission (now: Appendix H.1) presents the results of an experiment comparing our method with an oracle approach that uses the true latent variable $Z$ directly instead of learning a representation in the first stage. This comparison indicates that for lower-dimensional $A$, the error introduced by estimating the true latent is relatively mild, while this error becomes more pronounced when the dimension increases. We have now also included the oracle baseline in Figure 3 to visually demonstrate that the discrepancy is small in the one-dimensional case.

---

> ### Author Response · Authors · 2023-11-20
> **Response to Reviewer dGv9 (continued)**
>
> 6. > I find the sudden change of notation in Theorem 4 confusing.
>
>     We discussed this point at length among ourselves before submission. In the end, we concluded that highlighting the dependence on the observed distribution was worth the additional notational overhead. We have however now added “(e.g., (11) instead of (8))” in Footnote 3 to be more concrete about this.
>
> 7. > Can there be randomness in X, i.e., can X be a noisy observation
>
>     We have conducted an additional experiment in which we introduce random noise into $X$. The results indicate that our method is robust to the presence of noise in $X$ (when the noise is not too large). The results can now be found in Appendix H.3. Additionally, we have noted in the conclusion that future work could extend our theoretical results to cover this case.
>
> 8. > On page 2, the authors wrote "we allow for potentially unobserved confounders between Y and Z". How could there be confounding when the action is exogenous?
>
>     We allow for confounding between the latent $Z$ and the response $Y$ in Setting 1 because we do not assume that $U$ and $V$ are independent.
>
> 9. > Although the goal is very different, the approach reminds me of the negative control literature. The authors may want to discuss the connection.
>
>     We do not see a clear connection between the negative control literature and our work but we are happy to hear about any concrete connections you may have in mind.

---

### Official Review · Reviewer_k2zA · 2023-10-31

**Soundness:** 3 good
**Presentation:** 3 good
**Contribution:** 3 good
**Rating:** 8
**Confidence:** 4

**Summary:**

**Post rebuttal update**: I'd really like to see a couple of real-world examples that might satisfy the assumptions, but my other concerns are largely addressed. I raise my score to 8 accordingly.

The paper considers the causal effect of an intervention $A=a^*$, where value $a^*$ is not observed in the data; A affects the outcome Y through unobserved variable Z, and covariates X relates to Z through an injective function g. The paper uses a conditional moment restriction (CMR) implemented by a kernel method called maximum moment restriction (MMR), and it then uses the control function (CF) approach to identify the outcome function between Z and Y. Both the CMR and CF depend on the assumption that A affects Z linearly. The approach is theoretically guaranteed, and experiments on synthetic datasets support the theoretical analysis.

**Strengths:**

Using CF to achieve intervention extrapolation is a nice idea.

The theoretical analysis is serious and detailed (but I did not check the proofs in Appendix).

The paper is quite well written.

**Weaknesses:**

*Technical novelties seem to be weak*. Theorem 4 seems to be an adaptation of the CF approach in (Newey et al., 1999), and Theorem 6 seems to be an adaptation of the IV approach in (D’Haultfoeuille, 2011). If there are some technical novelties, they should be discussed and compared to the original works; otherwise, I suggest being more explicit about this weakness.

*Some assumptions are strong*; particularly, the linear model between Z and A, and the injective model and noiseless model between Z and X. Intuitively, X is an observable proxy of the hidden Z, and the assumption means there is no information loss in this proxy, which is strong. Moreover,  both assumptions involve hidden variable Z and add difficulty to practical judgments. Since both assumptions are inherent in the current approach, I do not expect the author(s) to address this weakness in the rebuttal, but the following could certainly be done.

*Discussion of the setting and comparison to related work*. The discussion of the relationship to reinforcement learning is interesting but does not touch on when can we possibly expect linearity and noiseless injectivity. It would be more interesting to draw and discuss a couple of real-world problems that might satisfy the assumptions.

On the other hand, (Khemakhem et al., 2020) and [1, 2], which are based on the former, are important related work that needs more discussions, and this would clarify the current approach.  For example, (Khemakhem et al., 2020) recover Z based on exactly the same graph as A→Z→X, and also assume g is injective but *allows an additive noise on X*; the identification also relies on assumptions on the A→Z part, where p(Z|A) is assumed to be an exponential family distribution but *allows nonlinearity*. Overall, I do not think the assumptions in (Khemakhem et al., 2020) are clearly stronger than those in the current work. Further, [1, 2] uses (Khemakhem et al., 2020) to estimate treatment effects, though not considering intervention extrapolation, [2] mentioned the ideas of CMR and CF in Sec 4.4.

[1] Wu, Pengzhou Abel, and Kenji Fukumizu. "$\beta $-Intact-VAE: Identifying and Estimating Causal Effects under Limited Overlap." International Conference on Learning Representations. 2022.
[2] Wu, Pengzhou and Kenji Fukumizu. Towards Principled Causal Effect Estimation by Deep Identifiable Models”. In: arXiv preprint arXiv:2109.15062 2021

*Additional experiments could be added*.

I think the ability to deal with unobserved confounders is a strength of the work. So, why not add experiments on this? I know eq24 contains hidden confounding, but this direction is not examined, e.g., by adjusting the strength of confounding and comparing to other methods.

As indicated above, adding real-world problems in experiments can greatly strengthen the work. Also, it would be interesting to replace the CMR part with iVAE (Khemakhem et al., 2020) and see how the results would change.

I will read the rebuttal and revised paper and raise my score to 8 if the issues/questions above are addressed.

**Questions:**

I cannot understand the importance of Proposition 3, and it seems just a trivial restatement of Def 2 and adds confusion to me.

I think the title would be better to stress the CF approach because this arguably contributes more to intervention extrapolation than “identifying representation”.

It is weird that Wiener’s Tauberian theorem is mentioned in the Abstract but not the main text.

---

> ### Author Response · Authors · 2023-11-20
> **Response to Reviewer k2zA**
>
> We thank the reviewer for the detailed feedback.
>
> We address the points raised in the 'Weaknesses' and 'Questions' sections below.
>
> 1. > Technical novelties seem to be weak. Theorem 4 seems to be an adaptation of the CF approach in (Newey et al., 1999), and Theorem 6 seems to be an adaptation of the IV approach in (D’Haultfoeuille, 2011). If there are some technical novelties, they should be discussed and compared to the original works; otherwise, I suggest being more explicit about this weakness.
>
>    Theorem 4 extends beyond the result of Newey et al. (1999) by not assuming that the latent variable $Z$ is observed. We believe the use of CFs in extrapolation is novel as well. As for Theorem 6, while part of the proof strategy is adapted from D’Haultfoeuille (2011) (as mentioned in the proof), the statement of the theorem itself is different. The key novelty lies in the use of the linear invariance condition for the identification of the hidden representation, which, to the best of our knowledge, has not been discussed in any prior work.
>
> 2. > Some assumptions are strong; particularly, the linear model between Z and A, and the injective model and noiseless model between Z and X. Intuitively, X is an observable proxy of the hidden Z, and the assumption means there is no information loss in this proxy, which is strong. Moreover, both assumptions involve hidden variable Z and add difficulty to practical judgments. Since both assumptions are inherent in the current approach, I do not expect the author(s) to address this weakness in the rebuttal, but the following could certainly be done.
>
>     > Discussion of the setting and comparison to related work. The discussion of the relationship to reinforcement learning is interesting but does not touch on when can we possibly expect linearity and noiseless injectivity. It would be more interesting to draw and discuss a couple of real-world problems that might satisfy the assumptions.
>
>     > As indicated above, adding real-world problems in experiments can greatly strengthen the work.
>
>      * We have now added Remark 6 in Appendix B (and have referenced this appendix after Setting 1) to discuss key assumptions in Setting 1 and whether they can be extended. Some extensions are straightforward; for example, one can relax the linearity assumption from $A$ to $Z$ if there is a known nonlinear transformation $h$ such that $Z := M_0 h(\tilde A) + V$; in this case, we use $A := h(\tilde A)$ and obtain a model that lies in our model class.  Furthermore, our setting accommodates any invertible nonlinear transformation on $Z$, too. To see this, if there is a nonlinear injective function $h$ such that $\tilde{Z} := h(M_0 A + V)$ and $X := g_0(\tilde{Z})$, we can define $Z := M_0 A + V$ and $X := (g_0 \circ h)(Z)$ and obtain a model in our model class. But key to our approach is that the model of $Z$ on $A$ has the ability to extrapolate, otherwise non-linear extrapolation for $\mathbb{E}[Y|\text{do}(A:=a^*)]$ is not possible.
>
>    * Motivated by your (and the other reviewers’) comments on noiseless injectivity, we have conducted an additional synthetic experiment, where we introduce random noise in $X$. The results indicate that our method is robust to a certain degree of noise in $X$. The results can now be found in Appendix H.3.
>    * We view our work as establishing a foundation for learning an identifiable representation with the goal of extrapolation. We hope that our work will spark further research in this direction which hopefully further relaxes assumptions and proposes alternative methodology to solve the task, with more flexible applicability to real-world problems.

---

> ### Author Response · Authors · 2023-11-20
> **Response to Reviewer k2zA (continued)**
>
> 3. > On the other hand, (Khemakhem et al., 2020) and [1, 2], which are based on the former, are important related work that needs more discussions, and this would clarify the current approach. For example, (Khemakhem et al., 2020) recover Z based on exactly the same graph as A→Z→X, and also assume g is injective but allows an additive noise on X; the identification also relies on assumptions on the A→Z part, where p(Z|A) is assumed to be an exponential family distribution but allows nonlinearity. Overall, I do not think the assumptions in (Khemakhem et al., 2020) are clearly stronger than those in the current work. Further, [1, 2] uses (Khemakhem et al., 2020) to estimate treatment effects, though not considering intervention extrapolation, [2] mentioned the ideas of CMR and CF in Sec 4.4.
>
>     We agree that our actions $A$ could be considered auxiliary variables. However, the work by Khemakhem et al., 2020 differs from ours in two important ways.
>     * Assumptions: one of the key assumptions in their setting is that the density $p(z|a)$ is conditionally factorized, meaning that the components of $Z$ are independent when conditioned on $A$. In contrast, our approach permits dependence among the components of $Z$ even when conditioned on $A$ (because in our setting, the components of $V$ can have arbitrary dependencies).
>     * Type of identifiability: maybe even more importantly, Khemakhem et al. provide identifiability up to point-wise nonlinearities which is not sufficient for intervention extrapolation. The main focus of our work is to provide an identification that facilitates a solution to the task of intervention extrapolation.
>
>     We have now included a discussion on the relationship to nonlinear ICA in Appendix A and have referenced this appendix in the related work section. Additionally, we have cited (Wu & Fukumizu, 2022) (along with other works suggested by another reviewer) in the introduction as an example of prior work that demonstrates the benefits of identifiable representations. Lastly, given the difference in the type of identifiability, we believe that an empirical comparison to iVAE may not be relevant.
>
> 4. > I think the ability to deal with unobserved confounders is a strength of the work. So, why not add experiments on this? I know eq24 contains hidden confounding, but this direction is not examined, e.g., by adjusting the strength of confounding and comparing to other methods.
>
>     Thank you for your suggestion. We have carried out an additional experiment to explore the impact of unobserved confounders on our approach's extrapolation performance. The results are now available in Appendix H.2. In short, our approach maintains consistent extrapolation performance across all levels of confounding.
>
> 5. > I cannot understand the importance of Proposition 3, and it seems just a trivial restatement of Def 2 and adds confusion to me.
>
>     Proposition 3 offers an additional perspective on Definition 3, specifically that $Z$ can be reconstructed from $\phi(X)$. Its statement is used in the evaluations of the experiments. We have now moved Proposition 3 to Appendix C.2.
>
> 6. > I think the title would be better to stress the CF approach because this arguably contributes more to intervention extrapolation than “identifying representation”.
>
>     We thought about including the control function approach in the title too, but it was challenging to find a title that emphasizes the main topic of learning representation for extrapolation while mentioning control functions, without making it overly complex or too long. We still have not found a title that we believe is better than the current one but we would certainly appreciate suggestions if you have any.
>
> 7. > It is weird that Wiener’s Tauberian theorem is mentioned in the Abstract but not the main text.
>
>     We agree that it is confusing to mention it in the abstract but not in the main text. Based on all reviewers’ comments, we have added several points to the main text, so space became a limiting factor. We did not find an explanation of the Tauberian theorem in the main text as important as the other points, so, we have now removed the reference to Wiener Tauberian theorem from the abstract. Its role becomes clear anyways, when reading the proof.

---

> > ### Comment · Reviewer_k2zA · 2023-11-21
> >
> > Thanks for the rebuttal. I'd really like to see a couple of real-world examples that might satisfy the assumptions, but my other concerns are largely addressed. I raise my score to 8 accordingly.
> >
> > Nice work!

---

### Official Review · Reviewer_iVVC · 2023-11-01

**Soundness:** 3 good
**Presentation:** 3 good
**Contribution:** 4 excellent
**Rating:** 8
**Confidence:** 4

**Summary:**

The authors propose a method to learn an effect through learning a representation of latents (Z) from observed data (X) where X and Z are related through some injective function (in practice an encoder model). It is assumed that interventions in A act on Z through which they act on some required outcome variable Y through linear functions. In such a setting the authors propose a method for learning the effects of ood interventions a^* in A. They show through experiments the efficacy of their method.

**Strengths:**

- Important problem. Can be thought of as OOD estimation of intervention effects through learning latent representations.
- Extremely well written paper!
- Crucially E[Y | do(A=a')] \neq E[Y | A=a'] for a' not in support of A.
- The propositions are exactly at the places that the reader thinks about the question, and are easily understandeable.
- The proofs of extrapolation are not straightforward. There have been several causal tools brought together to show the validity of extrapolation (invariance principle, mixing-unmixing, instrumental variable approaches. I quite liked the work.

**Weaknesses:**

- I did not see any major weakness. One model assumption that could be weakened in future work is the linearity assumption.
- The role of the Wiener’s Tauberian theorem in the proof of hidden representation being identifiable, upto affine transformation, is not clear to me. Since this has been claimed in the abstract it would be helpful to delineate where it has been used.

**Questions:**

- Proposition 1: If for all a in the support of A, E^S1[Y|A=a]=E^S2[Y|A=a], then how is there a set with positive lebegue measure over the support s.t the do distributions are not equal? The issue is one of measure of sets outside support being 0. Some clarification remarks would be helpful as to what positive measure outside supp(A) means.
- The linear assumption is reasonably strong. Future work may be required to extend it to GLM's or non-linear representation learning.
- Is the functional form of the SCM necessary for extrapolation? Can there be such analyses on CBNs?

---

> ### Author Response · Authors · 2023-11-20
> **Response to Reviewer iVVC**
>
> We thank the reviewer for the positive feedback.
>
> We address the points raised in the 'Weaknesses' and 'Questions' sections below.
>
> 1. > One model assumption that could be weakened in future work is the linearity assumption.
>
>     > The linear assumption is reasonably strong. Future work may be required to extend it to GLM's or non-linear representation learning.
>
>      We have now added Remark 6 in Appendix B (and have referenced this appendix after Setting 1) to discuss key assumptions in Setting 1 and whether they can be extended. Some extensions are straightforward; for example,  one can relax the linearity assumption from $A$ to $Z$ if there is a known nonlinear transformation $h$ such that $Z := M_0 h(\tilde A) + V$; in this case, we use $A := h(\tilde A)$ and obtain a model that lies in our model class.  Furthermore, our setting accommodates any invertible nonlinear transformation on $Z$, too. To see this, if there is a nonlinear injective function $h$ such that $\tilde{Z} := h(M_0 A + V)$ and $X := g_0(\tilde{Z})$, we can define $Z := M_0 A + V$ and $X := (g_0 \circ h)(Z)$ and obtain a model in our model class. But key to our approach is that the model of $Z$ on $A$ has the ability to extrapolate, otherwise non-linear extrapolation for $\mathbb{E}[Y|\text{do}(A:=a^*)]$ is not possible.
>
> 3. > The role of the Wiener’s Tauberian theorem in the proof of hidden representation being identifiable, upto affine transformation, is not clear to me. Since this has been claimed in the abstract it would be helpful to delineate where it has been used.
>
>     Thank you for the comment, we agree that it is confusing to mention it in the abstract but not in the main text. Based on all reviewers’ comments, we have added several points to the main text, so space became a limiting factor. We did not find an explanation of the Tauberian theorem in the main text as important as the other points, so, we have now removed the reference to Wiener Tauberian theorem from the abstract. Its role becomes clear anyways, when reading the proof.
>
> 2. > Proposition 1: If for all a in the support of A, E^S1[Y|A=a]=E^S2[Y|A=a], then how is there a set with positive lebegue measure over the support s.t the do distributions are not equal? The issue is one of measure of sets outside support being 0. Some clarification remarks would be helpful as to what positive measure outside supp(A) means.
>
>     Please note that we refer to positive measure with respect to Lebesgue and not with respect to $\mathbb{P}_A$. The existence of such a set $\mathcal{B}$ is possible if $\mathcal{A} \setminus \text{supp}(A)$  has positive Lebesgue measure (and, indeed, the set $\mathcal{B}$ cannot be a subset of $\text{supp}(A)$, it must be a subset of $\mathcal{A} \setminus \text{supp}(A)$). We have now added clarifying comments to avoid possible confusion for future readers.
>
> 3. > Is the functional form of the SCM necessary for extrapolation? Can there be such analyses on CBNs?
>
>     Yes, the functional form of the SCM is necessary for extrapolation. As far as we can see, describing the data-generating process with a causal Bayesian network alone would not permit specific model specifications, such as the linearity of $Z$ on $A$ or the additivity of $V$ on $Z$.

---

### Author Response · Authors · 2023-11-20
**General response to all reviewers**

We thank all the reviewers for taking the time to review our manuscript; we highly appreciate the positive comments and the detailed feedback.

We have submitted a revised version of the manuscript where we have highlighted important changes in red.

---

### Meta-Review · Area_Chair_6H4c · 2023-12-11

**Metareview:**

In this paper, the authors discussed the causal effect estimation from a representation view, under some structure causal dependency assumption.

During the rebuttal period, the authors successfully addressed most questions raised by reviewers. All the reviewers acknowledge the novelty and the clarity of the paper. This paper is a clear acceptance.

**Justification For Why Not Higher Score:**

However, there are still quite a few questions, including

1, more detailed discussion and comparison w.r.t. the existing literature;

2, non-verifiability of the linear dependency and the structure assumptions. (I understand this might not specific problem of this paper, but definitely should be considered in future work. )

**Justification For Why Not Lower Score:**

All the reviewers acknowledge the novelty in causal community and the clarity of the paper. This paper is a clear acceptance.

---

### Decision · Program_Chairs · 2024-01-16

Accept (poster)